# On the molecular origins of the ferroelectric splay nematic phase

Richard J. Mandle[1,2✉], Nerea Sebastián[3], Josu Martinez-Perdiguero[4] & Alenka Mertelj[3✉]

Nematic liquid crystals have been known for more than a century, but it was not until the 60s–70s that, with the development of room temperature nematics, they became widely used in applications. Polar nematic phases have been long-time predicted, but have only been experimentally realized recently. Synthesis of materials with nematic polar ordering at room temperature is certainly challenging and requires a deep understanding of its formation mechanisms, presently lacking. Here, we compare two materials of similar chemical structure and demonstrate that just a subtle change in the molecular structure enables denser packing of the molecules when they exhibit polar order, which shows that reduction of excluded volume is in the origin of the polar nematic phase. Additionally, we propose that molecular dynamics simulations are potent tools for molecular design in order to predict, identify and design materials showing the polar nematic phase and its precursor nematic phases.

[1] School of Physics and Astronomy, University of Leeds, Leeds, UK. [2] Department of Chemistry, University of York, York, UK. [3] Jožef Stefan Institute, Ljubljana, Slovenia. [4] Department of Physics, University of the Basque Country (UPV/EHU), Bilbao, Spain. ✉email: r.mandle@leeds.ac.uk; alenka.mertelj@ijs.si

Ferroelectric, ferromagnetic and ferroelastic materials hold a privileged position in material sciences and condensed matter physics research. Their remarkable properties (e.g. shape memory effect, pyroelectric effect, large mechanical, magnetic and electric susceptibilities) demonstrate their potential for use in a wide range of technologies. Although there is no fundamental reason preventing ferromagnetic or ferroelectric order in dipolar liquids, i.e., liquids made of constituents with dipole moment, it was not until very recently, that three different liquid systems with ferroic order have been realized experimentally[1–3]. Because dipolar interactions are strongly anisotropic, suitable (anisotropic) positional correlations between nearest neighbours are crucial for the appearance of ferroic order[4]. In a liquid, in which the constituents move and rotate, the positional correlations are short-ranged—they span to a few nearest neighbours, and they strongly depend on the shape of the constituents. It seems that the deciding factor for the appearance of long-range ferroic order is such anisometric shape of the constituents which also leads to the formation of anisotropic liquids, better known as liquid crystals (LCs). In the case of a simple uniaxial nematic phase (N), the anisotropic constituents are on average oriented in the same direction, denoted by a unit vector—director. However, for a long time, the realization of ferroic nematic phases remained elusive. Not long ago, it has been shown that, as theoretically predicted[5], disk-like shape can lead to uniform ferromagnetic nematic order in colloids of disk-like magnetic particles dispersed both in isotropic liquids[2] and in nematic LCs (NLC)[1,6], and to helical magnetic order when dispersed in chiral NLC[7,8]. Still, polar nematic phases on bulk molecular materials seemed to be prevented due to entropic and dipolar interactions between constituting molecules.

Recently, it has been shown that some materials made of elongated molecules, with large longitudinal electric dipole moment and a side group that gives molecules a slight wedge shape, exhibit two nematic phases[9,10]. The high-temperature phase is an apolar uniaxial nematic (N), the kind widely exploited in current LCD technology, while the low-temperature phase exhibits ferroelectric ordering on the macroscopic scale of several microns[3]. Because molecular shape lacks head-tail symmetry, polar ordering of the molecules causes orientational elastic instability. The transition between the phases is a ferroelectric–ferroelastic transition[3], in which a divergent susceptibility, typical of a ferroelectric transition, is accompanied by the softening of the splay elastic constant which causes the low-temperature ferroelectric phase to be non-uniform[3,11], i.e. ferroelectric splay-nematic phase ($N_S$). As it is impossible to fill the space with homogenous Japanese fan-like splay deformation, the exact structure of the director field in the $N_S$ will depend on the confinement conditions (size, shape and boundary condition of the container), the orientational elastic constants, the electric polarization and ion concentration. The structure can exhibit 1D[12,13] or 2D[14] modulation, regular or irregular defect lattice[15] or the splay can be combined with other types of deformation, e.g. the twist deformation[16]. Experimentally, a modulated structure with microns size periodicity has been observed[3,11]. It is anticipated that when the material is confined to a layer with a thickness comparable to the periodicity, more metastable structures can occur[17]. The high saturated electric polarization values measured in the typical material RM734[18] show that materials exhibiting the $N_S$ phase are promising for a variety of applications such as low power fast electro-optic switching devices as well as LC-based photonic technologies, e.g. switchable optical frequency converters. Up to now examples of materials exhibiting the $N_S$ phase are scarce and limited to some RM734 analogues[9]. So far, it has been shown that for RM734-like materials, a short terminal chain (OMe, OEt) coupled with a lateral group (OMe, OEt or

OPr) is prerequisite for the formation of the $N_S$ phase, as is a terminal nitro group. The reasons for the dependency of the ferroelectric $N_S$ phase upon these structural features are not clear; understanding them is clearly a significant barrier to developing improved materials, which may eventually be deployed into applications. Also, an LC compound with a 1,3-dioxane unit (named DIO) was reported to exhibit similar textures, very large dielectric anisotropy and polarity in a low-temperature nematic phase[19]. More recently, Li et al. have reported several materials, with compatible molecular structures, which exhibit the $N_S$ phase and demonstrated that the polar nematic phase of DIO is the same as that shown by RM734[20]. Unfortunately, so far the $N_S$ phase has only been observed at high temperatures creating a pressing need for the development of materials which would exhibit the ferroelectric nematic phase at much lower temperatures, suitable for applications. Understanding the microscopic mechanism which leads to the formation of the ferroelectric phase is therefore imperative; such knowledge, combined with the development of predictive algorithms using computational models, would help to tailor the design of materials that exhibit this phase.

It has been shown that when the nitro group of RM734 is replaced by a nitrile group (RM734-CN), the resulting material does not exhibit the $N_S$ phase[9]. Notably, binary mixtures of RM734 with RM734-CN do not show the $N_S$ phase either; even with as little as 10 wt% of RM734-CN, the $N_S$ phase is suppressed entirely.

In this paper, we aim to gain insight into the driving mechanism for the formation of the $N_S$ phase, by carrying out a comprehensive comparative study of both materials, RM734 and RM734-CN. Experimentally, we focused on those properties that seem to be critical in the N–$N_S$ transition, by performing dielectric spectroscopy, dynamic light scattering and WAXS/SAXS experiments. The differences between both materials are analysed in-depth via molecular dynamics simulations; enabling us to develop a model which is then successfully tested against three further variants of RM734, one of which also exhibits the $N_S$ phase.

## Results

**Distinct dielectric relaxation spectra.** Molecular structure and reported transition temperatures[9] of RM734 and RM734-CN are given in Fig. 1. Detailed description of the phase behaviour can be found in the 'Methods' section and Supplementary Figs. 1 and 2. The dielectric spectrum of the N phase of RM734 is distinctive[3],

**Fig. 1 Chemical structure of mesogens used in this study.** Molecular structures and transition temperatures (°C) of **a** RM734—Cr 139.8 ($N_S$ 132.7) N 187.9 Iso, and **b** RM734-CN—Cr 173.2 N 200.4 Iso.

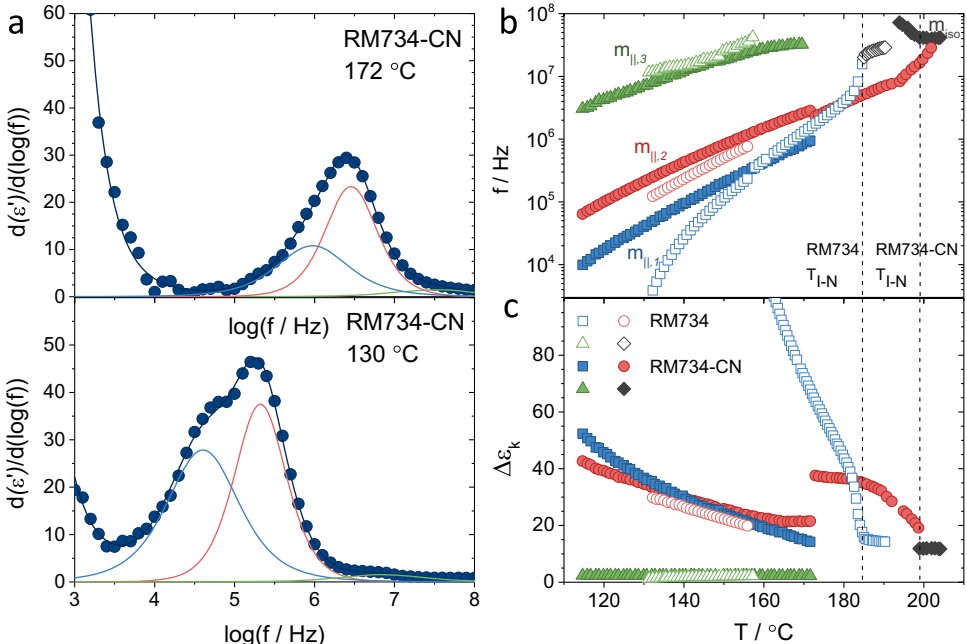

**Fig. 2 Broadband dielectric spectroscopy. a** Two examples at different temperatures of the derivative of the real part of the permittivity $d(\varepsilon')/dlog(f)$ (full circles) and the corresponding fit for RM734-CN (solid lines). On the right, temperature dependence of the **b** frequencies and **c** amplitudes of the characteristic dielectric relaxation processes for RM734 (empty symbols) (ref. [3]) and RM734-CN (full symbols) as obtained from fits to the Havriliak-Negami equation[59]. In both cases, three characteristic processes are observed. A high-frequency one $m_{\parallel,3}$ (green triangles), related to the fast molecular reorientation around the molecular long axis, and, unlike conventional nematics, two low-frequency processes: $m_{\parallel,2}$ (red circles), attributed to the molecular reorientation around the short molecular axis and $m_{\parallel,1}$ (blue squares) accounting for the increase of molecular orientation correlations.

being characterized by the gradual emergence of a relaxation mode, whose strength ($\Delta\varepsilon$) strongly increases and its relaxation frequency rapidly decreases when approaching the N–N$_S$ phase transition. Such behaviour is characteristic of increasingly cooperative dipole motions, evidencing the growth of polar correlations in the N phase. We have performed the same studies for the analogue RM734-CN material in the isotropic and nematic phase.

Similarly to RM734, RM734-CN spontaneously aligned homeotropically on the untreated gold electrode surfaces, and thus, we measured the parallel component of the permittivity. To ensure no director reorientation during the measurement, the oscillator level was set to 30 mV$_{rms}$ and measurements were performed on cooling. Temperature and frequency-dependent dielectric spectra in the nematic phase is given in Supplementary Note 2 The characteristic frequencies and amplitudes of the different relaxation modes obtained for RM734-CN ('Methods') are shown in Fig. 2 and compared to those already reported for RM734[3]. The isotropic phase is characterized by a broad single relaxation process (m$_{iso}$) with frequency around 40 MHz and amplitude $\Delta\varepsilon_{iso} \sim 12$. On cooling, immediately after the I–N transition, dielectric spectra are characterized by a relaxation mode at lower frequencies and with growing amplitude. This mode, although close to a Debye-type relaxation at high temperatures, slowly broadens on decreasing the temperature. This same behaviour was observed for RM734. As in the latter case, far below the transition, it becomes evident that the seemingly single relaxation mode detected at high-temperature range must in fact be deconvoluted into two different relaxation modes, m$_{\parallel,1}$ and m$_{\parallel,2}$. In line with the observations for RM734, the higher frequency one, m$_{\parallel,2}$, can be attributed to molecular rotations around the short molecular axis, while the lower frequency one, m$_{\parallel,1}$, would correspond to the collective reorientation of the dipole moments. However, in the case of RM734-CN, dipole

correlations are remarkably weaker than in the case of RM734. In contrast to the clear softening of the mode for RM734, the characteristic frequency of m$_{\parallel,1}$ for RM734-CN shows Arrhenius-like behaviour in the full N temperature range and its strength, although increasing, is far from the diverging trend of RM734. In the latter $\Delta\varepsilon_{\parallel,1}$ is much larger than $\Delta\varepsilon_{\parallel,2}$, and the spectrum is dominated by the collective mode. On the other hand, $\Delta\varepsilon_{\parallel,1}$ and $\Delta\varepsilon_{\parallel,2}$ in RM734-CN are comparable, indicating that collective reorientations, although present, are weaker. Noteworthy, the amplitudes of the molecular mode m$_{\parallel,2}$ are comparable in both materials, in agreement with the similar molecular dipole moments of both analogues (see the 'Electronic structure calculations' section). Description of the behaviour in the high-frequency range can be found in Supplementary Note 2.

**Viscoelastic properties in the N phase.** One more distinctive characteristic of the N–N$_S$ transition in RM734 is the remarkable pretransitional softening of the splay elastic constant[12]. The ground state of the classical apolar nematic phase is uniform, i.e., the orientation of **n** does not change with position. We measured the orientational elastic constants in RM734-CN associated with the increase of the energy due to the splay deformation of **n** using a combination of dielectric Frederiks transition and dynamic light scattering intensities (see 'Methods'[12]) and compared them to those of RM734[12]. At high temperatures, $K_1$ of both materials (Fig. 3a) are comparable and unusually low for a NLC. However, it immediately becomes evident that while $K_1$ of RM734-CN remains practically constant over the measured N range, in the case of RM734, the splay elastic constant strongly decreases as the N–N$_S$ transition is approached on cooling.

Similar contrasting behaviour is observed for the ratio $K_1/\eta_1$ obtained from the relaxation rates of the splay fluctuation modes (Fig. 3b). For RM734-CN the splay mode becomes slightly slower

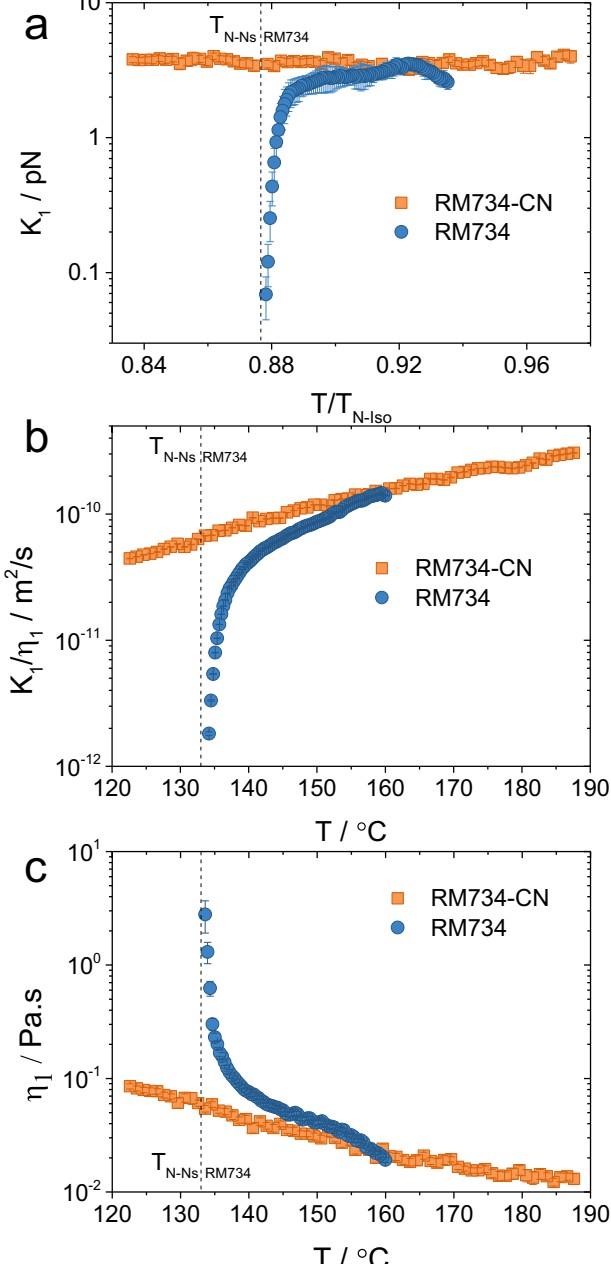

**Fig. 3 Viscoelastic properties of RM734 and RM734-CN in the nematic phase.** Temperature dependence of **a** the splay elastic constant $K_1$ **b** the ratios $K_1/\eta_1$ and **c** viscosities $\eta_1$ for RM734 (blue full circles) (ref. [13]) and RM734-CN (orange full squares). While values for $K_1$ are obtained from the combination of dynamic light scattering intensities of the splay orientational fluctuations and the Frederickz transition, diffusion coefficients $K_1/\eta_1$ are obtained from the relaxation rates. Combination of both yields values of the viscosities. Comparison of results for both materials shows that the substitution of the nitro group by a nitrile group has a strong effect on the viscoelastic properties, hindering the softening of the splay elastic constant. Vertical dashed lines mark the N–N$_S$ transition temperature $T_{N-N_S}$. Error bars correspond to s.d.

on lowering the temperature, while a strong softening is detected for RM734 on approaching the N–N$_S$ transition. From the splay elastic constant and the diffusion coefficients $K_1/\eta_1$ we obtained the temperature dependence of the splay viscosity (Fig. 3c), showing a steep increase before the N–N$_S$ transition for the *RM734* as opposed to the classical Arrhenius-like tendency observed for RM734-CN.

**Electronic structure calculations**. Thus far we have shown the dielectric and viscoelastic properties of RM734 and RM734-CN to be distinct; we therefore turned electronic structure calculations and molecular dynamic simulations in an attempt to understand the observed differences and how they originate in molecular structure.

An initial investigation into the conformational preference of RM734 and RM734-CN was made with DFT calculations. Starting from a geometry optimised at the M06HF-D3/aug-cc-pVTZ level, we performed fully relaxed scans about each of the dihedrals at the same level of theory; i.e. the geometry was optimised for each conformation sampled. Selected dihedrals are shown via the highlighting of their central bond in Fig. 4. Not unexpectedly, we find that the choice of nitro or cyano terminal unit does not affect the molecular electronic structure in such a way that the conformational preference is affected.

The calculated molecular dipole moments for the global energy minimum geometries of RM734 and RM734-CN, at the M06HF-D3/aug-cc-pVTZ level of DFT, are 11.4 and 11.2 Debye, respectively. For a molecule oriented with its mass inertia axis (taken to be the eigenvector associated with the smallest eigenvalue of the inertia tensor) along $x$, the dipole vector components (in Debye) are {10.9931,2.9740,0.4941} and {10.7360,2.9549,0.4850}, resulting in an angle between the direction of the dipoles and the molecular axis of 18.3° and 20° for RM734 and RM734-CN, respectively (Supplementary Fig. 8). The calculated eigenvalues of polarizabilities at 800 nm (given in Supplementary Table 1) at the same level of theory, show that while both molecules have comparable isotropic polarizabilities, the anisotropic polarizability is larger for RM734-CN in correspondence with the larger value of the birefringence measured for the nitrile analogue (Supplementary Fig. 6).

**Molecular dynamics simulations and X-ray scattering**. The X-ray scattering behaviour of RM734 (and that of other structurally related splay-nematic materials) is distinct from classical NLCs for two principal reasons: the scattering intensity is extremely weak in the liquid and liquid-crystalline states; multiple additional diffuse small-angle reflections (002, 003) are present in the N and N$_S$ phases[9]. Neither of these behaviours is observed for the analogous nitrile-terminated material, RM734-CN, or indeed conventional NLCs.

We conjecture that the occurrence of the N$_S$ phase and the observation of multiple small-angle X-ray scattering peaks may be related; specifically, that the additional diffuse small-angle reflections observed in the N phase of RM734 (and other materials) encode information about polar order, and so can be used to identify those N phases which potentially can lead to the formation of the N$_S$ phase. To demonstrate this, we sought to calculate/simulate X-ray scattering patterns for polar and apolar nematic states of both RM734 and RM734-CN.

Computation of X-ray scattering intensities for isolated molecules (e.g. those from DFT calculations) lacks a description of the nematic structure factor and so cannot explain the observed differences between RM734 and RM734-CN. We, therefore, turned to molecular dynamics (MD) simulations as a means to generate polar (parallel) and apolar (antiparallel) nematic phases comprised of RM734 and RM734-CN, with these then used for subsequent calculation of X-ray scattering intensities.

We performed MD simulations of both materials in both polar and apolar starting configurations, at a range of temperatures (see 'Methods'). We do not observe a splay modulated structure in our simulations (Fig. 5a); however, this is not unexpected given the modulation period of the N$_S$ phase has been measured to be on

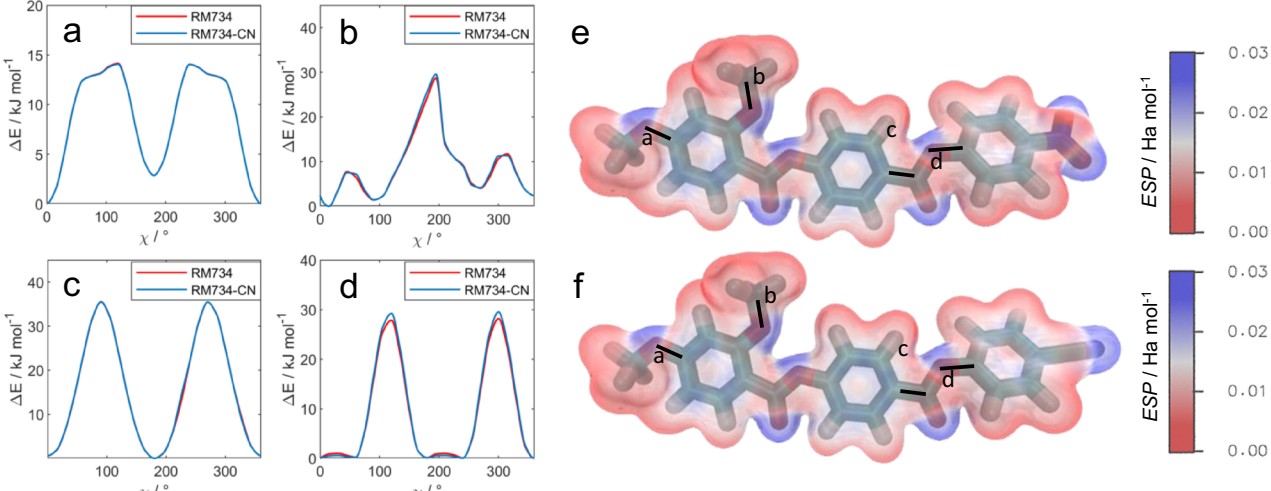

**Fig. 4 DFT conformational investigations. a–d** Conformational energy as a function of the rotational angle for the labelled bonds of RM734 (red solid lines) and RM734-CN (blue solid lines). Minimum energy geometries of RM734 (**e**) and RM734-CN (**f**), each with the electrostatic potential isosurface (isovalue 0.04). All calculations were performed at the M06HF-D3/aug-cc-pVTZ level of DFT, as implemented in Gaussian G09 revision d01.

the order of several microns[3,11], whereas the final dimensions of each simulation box are ~10 × 6 × 6 nm$^3$.

Calculated values of $P2$ (see 'Methods') as a function of temperature together with the obtained density values, the latter in particular offering a critical observation (Fig. 5b–d). At a given temperature, the density of RM734-CN is invariant in the polar and apolar nematic simulations (Fig. 5d and Supplementary Fig. 17). Conversely, we find that, at a given temperature, the density of RM734 is somewhat (~0.5%) larger in the polar nematic than in the apolar and simultaneously molecules exhibit larger translational diffusion constant in polar than in apolar state (Supplementary Table 3). The simulated density of RM734 is around 15% less than measured in the solid state by X-ray diffraction (1.473 g cm$^3$; see CCDC deposition number 1851381, Supplementary Note 12)[12].

For both we find $P2$ to be nematic like (i.e. $P2 \geq 0.3$) at and below 450 K, and isotropic (i.e. $P2 < 0.3$) at and above 500 K. This compares favourably with experimentally determined clearing points (RM734 = 461 K, RM734-CN = 473.6 K). The obtained values of $P2$ are compared with experimental values obtained via WAXS (Table 1) at 400 K. It is immediately apparent that both polar and apolar simulations yield a larger orientational order parameter than that obtained experimentally; simulation mean values of ~0.75 in all four cases compare with experimentally obtained values of 0.68 and 0.62 for RM734 and RM734-CN, respectively[21]. This overestimation is consistent with other fully atomistic MD simulations[22–24]. In all four simulations the biaxial order parameter, $B$, was negligible (lower than 0.05). As expected, the polar order parameter ($P1$) is large (~0.9) for simulations in the polar configuration and negligible (lower than 0.05) for apolar simulations. As molecular axis and dipole moment of each molecule are not parallel, we can also calculate the order parameter $P1$ of the polarization vector, i.e. $P1$(dipole), as the total dipole moment of the simulation box divided by the sum of the dipoles of the individual molecules (Table 1). Difference between $P1$ and $P1$(dipoles) arises from the fact that the molecular axis and the molecular dipole are not parallel. In addition, we can define the polarization vector ($P$) as the total dipole of the simulation box divided by its volume and calculate the angle between the director **n** and the direction of the polarization described by the unit vector **np**.

Briefly, comparing our MD simulations of RM734 with those reported in ref. [18], we find our simulations give near-identical values of $P2$ (polar = 0.787 ± 0.009, apolar = 0.78 ± 0.02, at 130 °C in ref. [18]). Values of the polar order parameter, $P1$ are also comparable (polar = 0.924 ± 0.003, apolar = 0.013 ± 0.004, at 130 °C) although our values are slightly lower for the polar simulation. In ref. [18], the simulated mass density is reported as 1.3 g cm$^3$ at 130 °C, but no distinction is made between the polar and apolar nematic states; this value is broadly comparable with our simulations of RM734. Important points of differentiation between this prior MD work and our own are the number of molecules per simulation (368 in ref. [18] versus 680 here), production MD simulation length about 20 ns in ref. [18] versus 250 ns here), and force field choice (polarizable APPLE&P in ref. [18], non-polarizable GAFF-LCFF here).

We calculate pair-correlation functions for RM734 and RM734-CN in both polar and apolar nematic states from the centre-of-mass of each molecule via a cylindrical binning procedure (Supplementary Fig. 13). Peaks indicate preferred pairing, and in each analysis steric repulsion between molecules leads to a region of low probability centred at $h,r = 0$ Å (Fig. 5e–h). RM734 in its polar configuration (Fig. 5e) displays large on-axis arcs at $h \approx \pm 20$ Å, which corresponds to head-to-tail pairing (e.g. terminal OMe to terminal $NO_2$); this feature is somewhat less pronounced in RM734-CN in the polar configuration (Fig. 5g), indicating a reduced preference for these configurations. The polar configuration of RM734 also exhibits several off-axis peaks which are absent in the apolar configuration and in RM734-CN in both polar and apolar configurations. In the polar configuration, RM734-CN has significantly stronger off-axis peaks at $h \approx r \approx 6$ Å than RM734; these corresponding to staggered parallel pairs of side-by-side molecules.

There is a subtler difference between the different packing modes in the apolar simulations (Fig. 5f and h), with a mixture of head-to-head and side-side pairs being common to both materials. The on-axis peaks at $h \approx \pm 20$ Å, for both materials, correspond to head-to-head (or tail-to-tail) configurations. The off-axis peaks at $h = r = 6$ Å correspond to staggered antiparallel pairs, which are known to be dominant for other low-molecular-weight nitrile-terminated materials (e.g. 5CB).

Our analysis of RM734 is in accord with the earlier simulation study by Chen et al.[18], and demonstrates that this material exhibits a strong preference for head-to-tail pairing. By also simulating RM734-CN, we find that the terminal nitrile is seemingly unable to sustain this pairing motif, instead adopting a staggered parallel

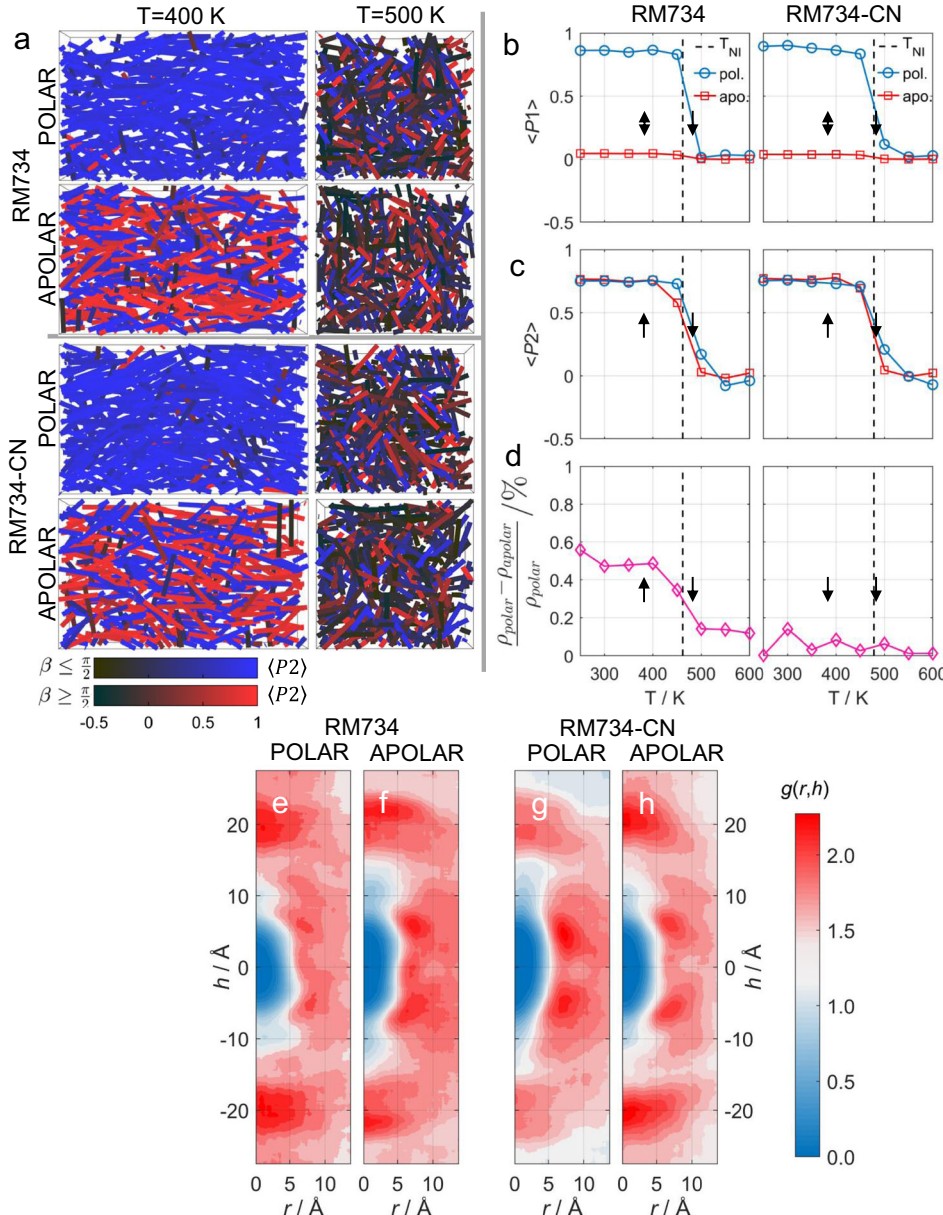

**Fig. 5 Molecular dynamics simulations. a** Snapshots of molecular dynamics simulations of RM734 and RM734-CN in polar and apolar configurations, and at temperatures of 400 and 500 K, respectively. In each plot, the coloured lines represent the molecular long axis. The order parameter of each molecule is represented by the colour depth of its corresponding line representation. Lastly, two colour scales are used: black to blue, where the molecular long axis makes an angle of ≤90° with the director, and black to red when this angle is ≥90° with the director. Each snapshot was taken at $t = 250$ ns. Plots of the order parameters $P1$ (**b**) and $P2$ (**c**) as a function of simulation temperature for RM734 and RM734-CN in polar and apolar configurations. Plots of the difference in mean density between the polar and apolar simulations of RM734 and RM734-CN as a function of temperature (**d**). In (**b–d**), data points represent mean values from simulations at the indicated temperature, whereas solid lines are a guide to the eye. **e–h** cylindrical pair-correlation functions obtained for MD simulations of RM734 (**e**, **f**) and RM734-CN (**g**, **h**) in polar (**e**, **g**) and apolar (**f**, **h**) nematic configurations, respectively, at 400 K. The cylindrical pair-correlation functions were computed over the same time window as was used in the calculation of X-ray scattering data, i.e. 200–280 ns.

pairing, driven by CN–C(O)O interactions, in polar simulations. We suggest that the head-to-tail pairing is presumably crucial to the formation of the polar $N_S$ phase under experimental conditions.

Having demonstrated our ability to generate polar and apolar nematic configurations of both materials, and given the distinctive X-ray scattering behaviour of RM734, we next calculated the two-dimensional X-ray scattering patterns from our MD simulations (Fig. 6), subjecting these to the same analysis as used previously for experimental data (see 'Methods'). Notably X-ray scattering intensities calculated for polar nematic simulations of both cases feature multiple low angle peaks, whereas the corresponding apolar

nematic simulations do not (Fig. 6e, f). The implication being that the presence of multiple low angle scattering peaks is a consequence of polar nematic order rather than differences in molecular structure. We compare this with experimental X-ray scattering data (Fig. 6): RM734 exhibits several low angle peaks in both the nematic and $N_S$ phases (as do structurally related materials reported previously[9,10], however, these additional low angle peaks are absent in scattering patterns obtained for RM734-CN, which exhibits a typical apolar nematic phase. These results indicate that small-angle X-ray scattering is perhaps a useful, albeit indirect, probe of polar order within nematic and polar NLC.

**Table 1 Calculated order parameters.**

|  | *P2* | *B* | *P1(n)* | *P1(dipoles)* | *P* (C/m²) | ∠ (n,np) (rad) |
|---|---|---|---|---|---|---|
| RM734 (POLAR MD) | 0.75 ± 0.005 | 0.036 ± 0.005 | 0.895 ± 0.004 | 0.860 ± 0.008 | 0.064 ± 0.003 | 0.017 ± 0.001 |
| RM734 (APOLAR MD) | 0.76 ± 0.010 | 0.044 ± 0.011 | 0.017 ± 0.003 | 0.042 ± 0.010 | 0.0017 ± 0.0002 | 0.052 ± 0.003 |
| RM734-CN (POLAR MD) | 0.73 ± 0.005 | 0.012 ± 0.006 | 0.897 ± 0.004 | 0.855 ± 0.007 | 0.052 ± 0.003 | 0.004 ± 0.0007 |
| RM734-CN (APOLAR MD) | 0.77 ± 0.005 | 0.012 ± 0.006 | 0.009 ± 0.003 | 0.022 ± 0.006 | 0.00061 ± 0.0001 | 0.036 ± 0.003 |
| RM734 (WAXS) | 0.68 | – | – | – | – | – |
| RM734-CN (WAXS) | 0.62 | – | – | – | – | – |

Second-rank orientational order parameter (*P2*), biaxial order parameter (*B*), polar order parameter (*P1*), order parameter of the polarization vector (*P1(dipoles)*), polarization vector (*P*) and angle between the direction of the polarization vector and the director (∠(**n,np**)) at 400 K for polar or apolar molecular dynamics simulations; all values are an average over each timestep in the production MD run (30–280 ns) as described in the text, with plus/minus values corresponding to one standard deviation from the mean. Experimental values *P2* obtained by WAXS as described in ref. [21].

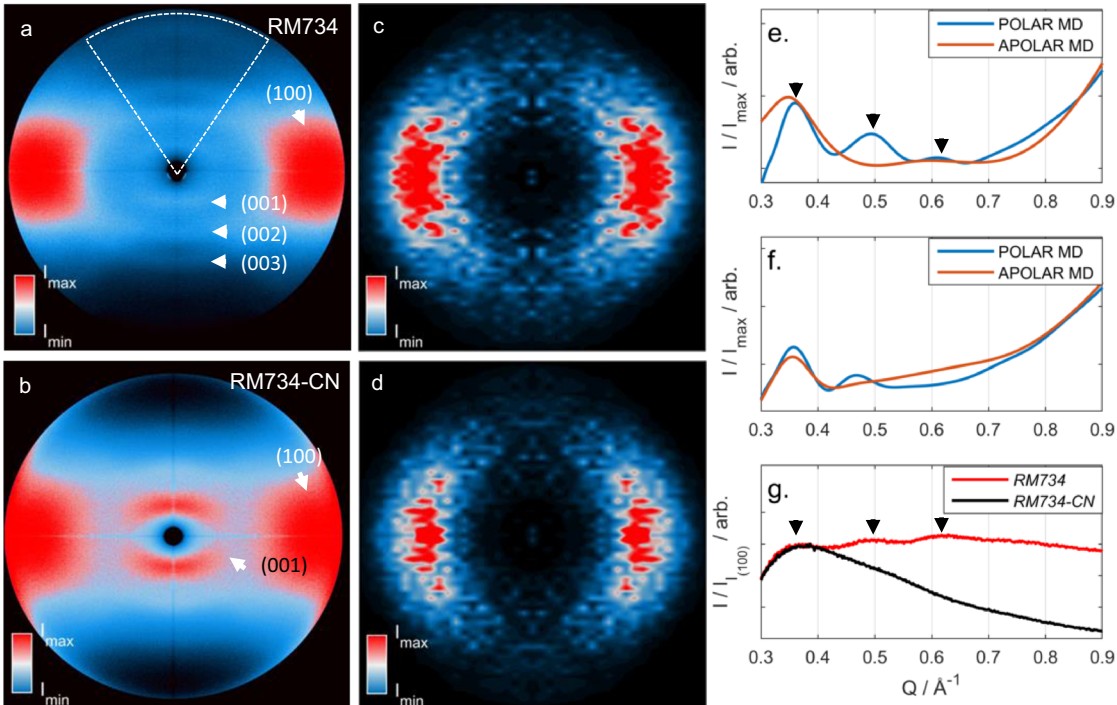

**Fig. 6 X-ray scattering.** Magnetically aligned two-dimensional X-ray scattering patterns for **a** RM734 and **b** RM734-CN in the nematic phase at 152 and 158 °C, respectively. Simulated 2D X-ray scattering patterns obtained from MD trajectories for RM734 with polar (**c**) and apolar (**d**) nematic organization, as described in the text. Calculated small-angle X-ray scattering intensities (Icalc) versus *Q* (Å⁻¹) for MD simulations with parallel and antiparallel dipole organisation for RM734 (**e**), and RM734-CN (**f**). **g** Experimental small-angle X-ray intensities (*I*/*I*max) versus *Q* (Å⁻¹) for RM734 (red solid line) and RM734-CN at 130° (black solid line). In (**a**), the dashed lines show the region that was radially integrated to obtain the data presented in (**g**); in (**e**) and (**g**), arrows are a guide to the eye and indicate the 001, 002 and 003 peaks.

## Discussion

Up to now, nematic phases have not shown long-range polar order, even when formed by rod-like molecules with large electric dipole moments. In a uniform nematic phase, random head/tail arrangement is entropically favourable, as well as from the dipole interaction point of view, because antiparallel orientation of neighbouring elongated dipoles is energetically favourable. RM734 demonstrated that a thermotropic polar nematic phase can be realized in a material made of slightly wedge-shaped molecules. However, the RM734-CN analogue reported here shows how a subtle chemical substitution completely modifies this phase behaviour. The first remarkable observation stemming from the comparison of both analogues is that, although splay elastic constant is low for RM734-CN, no pretransitional softening of it is detected as in the case of RM734. Another important observation arises from the evolution of the birefringence Δ*n* vs *P2* (Supplementary Fig. 6). Both materials have comparable polarizabilities, but noticeable different behaviour. The link between birefringence and *P2*, besides depending on the angular distribution function, is also influenced by the orientational molecular correlations[25]. This indicates that orientational molecular correlations differ in both materials. Such fact also relates to the dielectric behaviour of both analogues. While RM734 shows strong polar correlations culminating in the N–N$_S$ transition, the onset of such orientational correlations for RM734-CN, although discernable, is of much lower magnitude. The opposite difference is observed in terms of positional correlations between nearest neighbours. SAXS measurements evidence weaker positional correlations in the N phase of RM734 than in the case of RM734-CN, as inferred from the much weaker scattering intensity when compared to typical nematic scattering patterns. Comparison with RM734 might hinder the relevance of the findings for RM734-CN, however, physical properties of RM734-CN are far from being those of usual N phases. The splay elastic constant of RM734-CN is particularly low when compared with usual nematic materials as 5CB. The presence of two low-frequency

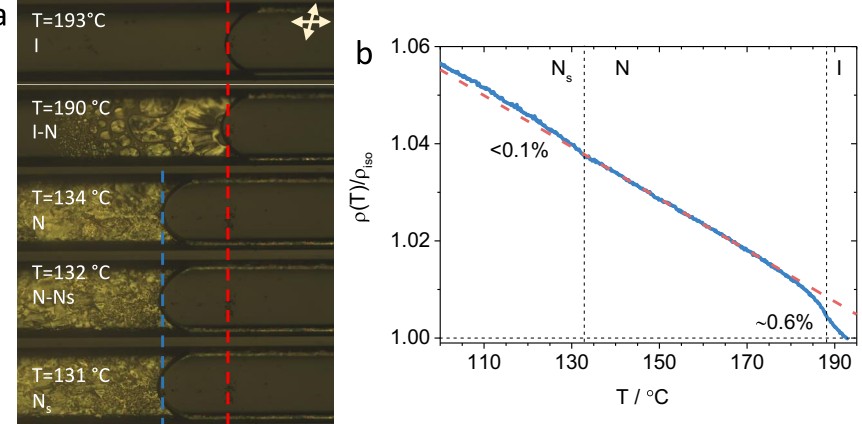

**Fig. 7 Density changes measurements. a** Polarizing optical microscopy snapshots at different temperatures of RM734 confined in a 500 × 50 capillary showing the volume changes as a function of temperature. Top-right corner double arrows indicate the directions of the slightly uncrossed polarizer and analyzer. Dashed red and blue lines are visual guidelines and mark the meniscus position in the isotropic phase and in the N phase before transition to $N_S$ phase, respectively. **b** Solid blue line gives the measured temperature dependence of the normalised density. Dashed line is the linear extrapolation from the data in the N phase.

relaxation modes, one arising from collective motions, clearly differs from the distinctive dielectric spectra of traditional nematic phases formed by rod-like molecules. For the latter, low-frequency spectra is governed by a single molecular relaxation mode associated with the reorientation of the molecular around their short axis. The great reported differences between the two very similar analogues raise the question of which are the underlying mechanisms driving the formation of the $N_S$ phase.

The density difference for the apolar and polar configurations of RM734, although small, can only be due to a more efficient packing of the molecules in the polar phase, i.e. polar order of the slightly wedge-shaped RM734 molecules, reduces the excluded volume, and, simultaneously, decreases attractive interaction energy (Supplementary Table 2). Such a difference is not observed for RM734-CN, which only shows the N phase. The dielectric data show that, although there are polar correlations in this material as well, they do not grow as temperature decreases. This suggests that the packing efficiency, which results in the reduction of the excluded volume and decrease of the attraction energy, is the underlying mechanism for the appearance of the polar nematic phase. Gregorio et al. showed that in a simple model of conical molecules, when density increases, polar order can build up causing the softening of $K_1$[26]. They state, that in their model, splay elastic constant shows a strong dependence on the molecular conical geometry. Inspired by this hypothesis, we measured the temperature dependence of relative changes of density for RM734 (see 'Methods'). Results show that below the N–$N_S$ transition there is a 0.1–0.2% density change between the extrapolated value for the N phase and the measured value for the $N_S$ phase (Fig. 7 and Supplementary Video 1). Such difference is smaller than that obtained from MD simulations (~0.5%), which is not surprising because polar order gradually grows already in the nematic phase and, at the weakly first-order phase transition, only part of the density difference occurs. Pair-correlation function analysis of MD simulations shows that the nitro group generates a distinct head-to-tail packing in the polar configuration which is not observed for RM734-CN. Instead, nitrile-terminated materials preferentially form a side-to-side packing, driven by the intermolecular interactions between the nitrile and ester groups, in the polar nematic configuration, and we suggest that due to the lateral methoxy units this arrangement has a larger excluded volume than the head-to-tail form preferred by the nitro-terminated material RM734.

We could further link MD simulations results to our experimental data. The additional small-angle X-ray scattering peaks observed for splay-nematic materials such as RM734 appear to be a consequence of the unique dipole ordering exhibited by these materials which, inter alia, leads to the formation of the $N_S$ phase and its remarkable properties. MD simulations of the polar state can faithfully reproduce these additional low angle reflections, whereas apolar nematic simulations cannot. Positional correlations are small, evidenced by the weak scattering signal, suggesting that some staggered positions are only slightly more favourable than others, which is further evidenced by the pair-correlation plots (Fig. 5e–h) and the radial distribution functions (RDFs) (Supplementary Figs. 14–15). Existence of the characteristic weak X-ray scattering pattern and the small angles peaks in RM734 already in the N phase far above the phase transition to the $N_S$ shows that local polar order is dominant already at high temperatures. The exceptional agreement between experimental and simulated X-ray scattering data shows that weak positional correlations and the appearance of additional peaks in the direction along the director may be used to predict and differentiate $N_S$ and precursor N phases from classical nematics.

Thus far we have restricted our atomistic MD simulations to just RM734 and RM734-CN, concluding with the hypothesis that the onset of polar order and the splay-nematic phase is driven by enhanced packing. We can further support this argument by simulating additional variants of RM734, shown in Fig. 8, in polar and apolar nematic configurations and looking for characteristic increased packing. We chose two materials which do not exhibit the $N_S$ phase: RM63 lacks the lateral methoxy group, whereas RM500 has a longer terminal alkyl chain. The synthesis and mesomorphic behaviour of both was reported previously[9]. We also studied an additional material, RM554, which exhibits the $N_S$ phase, and shows the same distinct X-ray scattering patterns as RM734 (Supplementary Fig. 18). RM554, possesses an additional fluoro substituent adjacent to the nitro group[9].

All simulations exhibit nematic order, as shown by the values of $P2$, and the biaxial order parameter is close to zero for all (Table 2). Whereas polar simulations give large values of $P1$, $P1$ (dipoles), and polarization vector $P$, these take near-zero values in the apolar simulations. The calculated density presents an important differentiation between materials that exhibit the $N_S$ phase (RM554) and those that do not (RM63, RM500). If we

make the argument that the splay-nematic phase is observed when the locally polar nematic configuration has a higher density than the equivalent apolar configuration, we begin to understand the molecular origins of this mesophase. For RM63 the calculated densities are effectively identical, and for RM500 the polar nematic is slightly lower in density than the apolar configuration. These results show that a lateral group is essential to give the packing advantage enjoyed by the polar configuration, however the presence of long terminal chains nullifies this advantage. These confinements to molecular structure present some impediments to the future engineering of materials for applications utilising the $N_S$ phase, which will presumably operate at ambient temperatures, however, this goes beyond the scope of this paper.

The calculated density of RM554 is significantly higher in the polar configuration than in the apolar nematic (~0.9%), which mirrors the behaviour seen for RM734. Notably, the increased dipole moment of the fluoro-substituted RM554, relative to the parent RM734, leads to a larger calculated polarisation vector ($P$) and, one expects, would lead to a larger measured spontaneous polarisation ($Ps$). This presents a route to future tuning of spontaneous polarisation of $N_S$ materials, with obvious ramifications for future applications. Pair-correlation analysis of an MD simulation of RM554 in the polar nematic configuration reveals the same head-to-tail packing mode observed for RM734 (Supplementary Fig. 18). Altogether, these observations highlight the possibility of using MD simulations as a predictive tool for

molecular design; by comparing densities of polar and non-polar configurations, through analysis of pair-correlation functions, in the pursuit of materials showing the polar nematic phase in the desired temperature range. GPU accelerated MD packages, and widespread availability of hardware, enable simulations with >100 ns day$^{-1}$ of performance for simulations featuring several tens of thousands of atoms (or more) that are required for stable liquid-crystalline order to emerge, raising the possibility of $N_S$ materials design guided by simulation.

A logical question to ask is why the polar and ferroelectric splay-nematic phase had not been observed previously given that it should be ubiquitous in rod-like molecules with large transverse electric polarity. This is more puzzling when we consider that such materials are widely employed in display technology, being studied en masse for over half a century. For display applications, materials must operate at ambient temperatures—this is typical achieved via mixture formulation of materials incorporating long (n-C$_3$H$_7$ to n-C$_9$H$_{19}$) terminal alkyl chains; such structural features are incompatible with the $N_S$ phase, as it is currently understood. Furthermore, in the context of materials for nematic LCDs, nitro substituents are generally inferior to nitriles, which themselves have been supplanted by multiply fluorinated materials[27]. At first glance, it may appear unusual that the $N_S$ phase had not been encountered until recently, but there is in fact far less overlap between the types of molecule studied so widely for display technology and those that exhibit this polar nematic variant than is first apparent.

Summarizing, although ferroelectric nematics have huge potential for applications, finding such materials has not been a trivial task as demonstrated by the long time it has taken to realize such phase of matter. Pairing optimized geometries and MD simulations of density and X-ray patterns, each on their own, might not be conclusive. However, altogether the knowledge we gain from MD simulations suggests that ferroelectric (splay) nematics could be designed in silico, rather than being discovered on an ad hoc basis, with the potential to accelerate applications utilizing this polar nematic molecular organization.

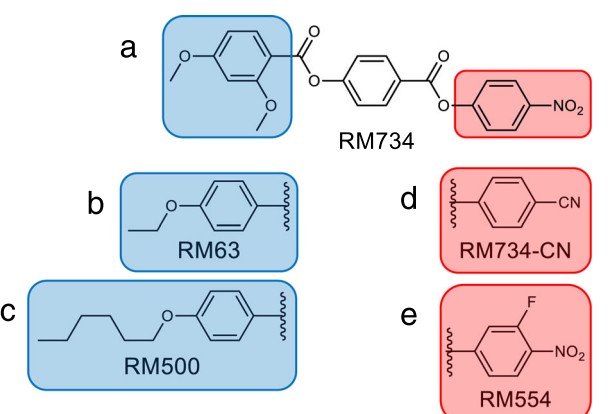

**Fig. 8 Additional structural variants of RM734 explored by MD in both polar and apolar configurations. a** RM734 and **d** RM734-CN compared in this work. **b** RM63, which compared to (**a**) RM734 lacks the lateral methoxy group and **c** RM500, with a longer terminal alkyl chain do not show $N_S$ phase. **e** RM554, an analogue of RM734 where a fluorine atom is positioned adjacent to the nitro group and whole terminal chain is lengthened by one carbon group, has been previously reported to also exhibit the $N_S$ phase[9].

## Methods

**Materials**. Both RM734 and RM734-CN were synthesised via literature methods[9]; their chemical structures and transition temperatures (°C) are given in Fig. 1, with monotropic phase transitions as reported by DSC measurements are presented in parenthesis ()[9]. All the experiments presented here have been performed on cooling. On cooling, RM734-CN shows an isotropic-nematic transition around 200 °C. However, the nematic phase can then be supercooled to temperatures around 120 °C even at cooling rates as slow as 0.25 °C/min.

On the other hand, RM734 shows a very rich phase behaviour. On heating from room temperature, a Cr–Cr (Cr I to Cr II) transition is observed between 80 and 90 °C depending on the heating rate (Supplementary Fig. 1). On further heating, the Cr II directly melts into the $N$ phase around 140 °C. Now, without heating further into the isotropic phase, cooling the sample reveals the following phase transition: N–132.7 °C–$N_S$–83 °C–Cr I (Supplementary Fig. 1b). The exact temperature at which the sample crystallizes depends on the cooling rate. Interestingly, depending on the history of the sample, the cooling rate and the cell surfaces, $N_S$ phase can be maintained down to room temperature (Supplementary

**Table 2 Calculated order parameters and density for the RM734 variants.**

|  | P2 | B | P1(n) | P1(dipoles) | P (C/m$^2$) | $\rho$ (g cm$^3$) |
|---|---|---|---|---|---|---|
| RM63 (polar) | 0.76 ± 0.010 | 0.017 ± 0.009 | 0.85 ± 0.06 | 0.83 ± 0.04 | 0.051 ± 0.002 | 1.260 ± 0.003 |
| RM63 (apolar) | 0.62 ± 0.021 | 0.026 ± 0.014 | 0.092 ± 0.009 | 0.068 ± 0.006 | 0.005 ± 0.0004 | 1.259 ± 0.003 |
| RM500 (polar) | 0.67 ± 0.034 | 0.064 ± 0.020 | 0.86 ± 0.08 | 0.84 ± 0.03 | 0.049 ± 0.003 | 1.175 ± 0.003 |
| RM500 (apolar) | 0.64 ± 0.024 | 0.073 ± 0.021 | 0.045 ± 0.011 | 0.022 ± 0.009 | 0.002 ± 0.0004 | 1.179 ± 0.003 |
| RM554 (polar) | 0.66 ± 0.015 | 0.027 ± 0.014 | 0.88 ± 0.09 | 0.87 ± 0.07 | 0.069 ± 0.0011 | 1.324 ± 0.003 |
| RM554 (apolar) | 0.68 ± 0.009 | 0.024 ± 0.008 | 0.061 ± 0.021 | 0.031 ± 0.011 | 0.009 ± 0.0007 | 1.311 ± 0.003 |

Second-rank orientational order parameter ($P2$), biaxial order parameter ($B$), polar order parameter ($P1$), order parameter of the polarization vector ($P1$(dipoles)), polarization vector ($P$) at 400 K for polar or apolar molecular dynamics simulations of homologous of RM734; all values are an average over each timestep in the production MD run (30–280 ns) as described in the text, with plus/minus values corresponding to one standard deviation from the mean.

Fig. 2) or a third Crystal phase can be obtained (Cr III) (Supplementary Fig. 1c). Subsequent heating from Cr III reveals that the $N_S$ phase can be obtained also on heating, which then transitions into the N phase around 130 °C. During this transition, some aligning features of the $N_S$ phase are retained which give rise to a further anchoring relaxation in the N phase. Details on these structural features are given elsewhere[17].

**X-ray scattering**. The X-ray scattering setup is described elsewhere, and datasets are reported elsewhere[9]. Values of Q were calibrated against a standard of silver behenate. The scattering pattern from an empty capillary (under the same experimental conditions as for each sample) was used as a background and was subtracted from each frame prior to analysis. Calculation of orientational order parameters from azimuthally integrated 2D WAXS patterns used the method described in refs. [28,29].

**Quantum chemical calculations**. Computational chemistry was performed in Gaussian G09 rev D01[30] on either the ARC3 machine at the University of Leeds, or using the same software package on the YARCC or Viking machines at the University of York. Output files were rendered using Qutemol[31], or VMD[32]. Calculations utilised the M06HF[33] hybrid DFT functional with additional D3 dispersion correction[34], and the aug-cc-pVTZ basis set[35]. The keywords Integral=UltraFine and SCF(maxcycles = 1024) were used to ensure convergence, while a frequency calculation was used to confirm the absence of imaginary frequencies and so confirm the optimised geometries were true minima.

Fully relaxed scans along the potential energy surface for a given dihedral were performed at the M06HF-D3/aug-cc-pVTZ level of DFT with a step size of 12° via the OPT = MODREDUNDANT keyword.

Polarizabilities and hyperpolarizabilities for RM734 and RM734-CN were calculated at the M06HF-D3/aug-cc-pVTZ level of DFT via the POLAR keyword, at frequencies of 400 and 800 nm (specified via CPHF = RdFreq).

**Molecular dynamics**. Fully atomistic molecular dynamics (MD) simulations were performed either in Gromacs 2019.3[36–42] on the ARC4 machine at the University of Leeds, with support for NVIDIA V100 GPUs through CUDA 10.1.168, or in Gromacs 2016.2 on the YARCC HPC at the University of York We used the General Amber Force Field (GAFF)[43] with modifications for liquid-crystalline molecules (GAFF-LCFF)[44], as this gives significant improvements over conventional GAFF in terms of predicting heats of formation and density, but also in term of its ability to generate thermodynamically stable liquid-crystalline order. Topologies were generated using AmberTools 16[45,46] and converted into Gromacs readable format with Acpype[47]. Atomic charges were determined using the RESP method[48] (see supplementary Note 14 and Supplementary Fig. 20) for geometries optimised at the B3LYP/6-31G(d) level of DFT[49,50] using the Gaussian G09 revision d01 software package[30].

We constructed initial low-density lattices of 680 molecules with random positional order. For polar nematic simulations, all molecular dipole moments were oriented along +x, while apolar nematic simulations have a 1:1 mixture of molecules aligned along +x and – x. Following energy minimisation, the simulation box was then compressed over 50 ps to a mass density of ~1 g cm$^3$ which is typical of that of low-molecular weight liquid crystals. Following compression, simulations were allowed to equilibrate for 30 ns, with subsequent production MD runs of a further 250 ns. No constraints were applied to enforce or preserve dipole order, which was allowed to evolve throughout the production MD run. Head-to-tail flipping of molecules is rarely seen in the nematic phase, as evidenced by the stable values of $P1 = \cos(\text{theta})$, where theta is the angle between a given molecular inertia axis and the nematic director (Supplementary Fig. 21). Temperature variations were studied by taking the final topology and trajectory of a completed simulation at 400 K, and heating (or cooling) to the desired temperature by setting *ref-t* to an appropriate value; we then performed a 30 ns equilibration followed by a further 250 ns production MD run. Analysis was performed during the time period 30–280 ns, i.e. the production MD run only. A timestep of 0.5 fs was used, and trajectories were recorded every 10 ps.

Simulations employed periodic boundary conditions in *xyz*. Bonds lengths were constrained to their equilibrium values with the LINCS algorithm[51]. System pressure was maintained at 1 bar using anisotropic Parrinello-Rahamn pressure coupling[52,53], enabling the relative box dimensions to independently vary in all dimensions. Compressibilities in *xyz* dimensions were set to 4.5e−5, with the off-diagonal compressibilities were set to zero to ensure the simulation box remained rectangular. Simulation temperature was controlled with a Nosé–Hoover thermostat[54,55]. Long-range electrostatic interactions were calculated using the Particle Mesh Ewald method with a cut-off value of 1.2 nm[56]. A van der Waals cut-off of 1.2 nm was used. MD trajectories were visualised using PyMOL 4.5.

We confirmed the nematic or isotropic nature of each simulation by calculating the second-rank orientational order parameter ($P2$) and biaxial order parameter ($B$) for all trajectory frames recorded during the production MD run, according to Eq. (1); $N$ is the total number of molecules (680 in all simulations), $m$ is the

molecule number within a given simulation.

$$Q_{\alpha\beta} = \frac{1}{N} \sum_{m-1}^{N} \frac{3 a_{m\alpha} a_{m\beta} - \delta_{\alpha\beta}}{2} \tag{1}$$

Where $\alpha$ and $\beta$ indicate two Cartesian directions, $a_m$ is the long axis of the mth molecule in the simulation, determined from its inertia tensor and $\delta_{\alpha\beta}$ is the Kronecker delta function. $Q_{\alpha\beta}$ is diagonalized to give three eigenvalues; the order parameter $P2$ corresponds to the largest eigenvalue of $Q_{\alpha\beta}$, and the biaxial order parameter $B$ corresponds to the difference between the two smallest eigenvalues. The polar order parameter, $P1$, was calculated according to:

$$P1 = \langle \cos\theta \rangle \tag{2}$$

where $\theta$ is the angle between the molecular axis and the local director.

High resolution one-dimensional SAXS curves ($I$ versus $Q$) were computed from MD trajectories using CRYSOL[57]. Simulated two-dimensional WAXS patterns were computed from MD trajectories using a modification of the procedure described by Coscia et al.[58]. Calculation of orientational order parameters from azimuthally integrated simulated 2D WAXS patterns used the method described in refs. [28,29].

For each simulation, we calculated two-dimensional WAXS patterns as an average of trajectories in the time window 200–280 ns (see Supplementary Note 9 and Supplementary Fig. 16). We next calculated small-angle X-ray scattering intensities as a function of Q every 0.5 s for frames in the time window 200–280 ns; values presented in Fig. 6e, f are an average of these 160 frames to remove any effect from instantaneous positional order.

**Broadband dielectric spectroscopy**. Spectroscopy measurements of the complex dielectric permittivity $\varepsilon(\omega) = \varepsilon'(\omega) - i\varepsilon''(\omega)$ were carried out in the $10^3 - 1.1 \times 10^8$ Hz frequency range with a HP4294 impedance analyzer. The material was placed in the nematic phase between two circular gold-plated brass electrodes (5 mm diameter) acting as a parallel-plate capacitor. The separation between electrodes was fixed by 50-µm-thick silica spacers. This sample was accommodated in a modified HP16091A coaxial fixture with a sliding short-circuit along its 7 mm coaxial line section and central conductor. The temperature of the sample was controlled down to ±50 mK in a Novocontrol cryostat. From the comparison with results obtained for the electrodes treated for homeotropic alignment, those obtained in classical glass ITO cells for parallel alignment, and observations of the orientation of the N phase when in direct contact with the ITO surface, we concluded that in the bare gold electrodes the sample spontaneously aligns homeotropically in the nematic phase. All measurements were performed on cooling from the isotropic phase.

At each temperature the characteristic frequency and amplitude of each relaxation process are obtained by fitting $\varepsilon(\omega)$ to the Havriliak-Negami equation[59]:

$$\varepsilon(\omega) = \sum_k \frac{\Delta\varepsilon}{\left[1 + (i\omega\tau_{\text{HN}})^{\alpha_k}\right]^{\beta_k}} + \varepsilon - i\frac{\sigma_0}{\omega\varepsilon_0} \tag{3}$$

Detailed description of the fits and examples can be found in Supplementary Note 2, Fig. 2 and Supplementary Fig. 4 for RM734-CN and Supplementary Fig. 5 for RM734 for completeness.

**Dynamic light scattering**. In the DLS experiments, we used a standard setup, using a frequency-doubled diode-pumped ND:YAG laser (532 nm, 80 mW), an ALV APD based "pseudo" cross-correlation detector, and ALV-6010/160 correlator to obtain the autocorrelation function of the scattered light intensity. The direction and the polarization of the incoming and detected light were chosen so that pure splay mode was observed[12]. A single mode optical fibre with a GRIN lens was used to collect the scattered light within one coherence area. We fitted the intensity autocorrelation function $g_2$ with $g_2 = 1 + 2(1 - j_d)j_d g_1 + j_d^2 g_1^2$, where $j_d$ is the ratio between the intensity of the light that is scattered inelastically and the total scattered intensity, and $g_1$ was a single exponential function, $g_1 = e^{-t/\tau}$. The relaxation rate $1/\tau$ was attributed to the splay eigenmode of orientational fluctuations with the wavevector $q$ equal to the scattering vector $q_s$. The scattered intensity of the mode was determined as a product $j_d I_{\text{tot}}$, where $I_{\text{tot}}$ was the total detected intensity. The scattered intensity from a pure splay mode is, $I_1 \propto \left(\Delta\varepsilon_{\text{opt}}\right)^2 / K_1 q^2$, and the relaxation rates $1/\tau_1 = K_1 q^2 / \eta_1$, where $q$ is the scattering vector[25]. The temperature dependence of the anisotropy of dielectric tensor at optical frequencies $\Delta\varepsilon_{\text{opt}}$ was obtained from measurements of $\Delta n$ (Supplementary Fig. 6). With this method, the temperature dependence of the splay elastic constant and viscosity are obtained, but not their absolute value. We determined the absolute values of $K_1$ and $K_3$ at 180 °C from capacitance measurement of the Fredericks transition, where the threshold voltage for the reorientation of the material in a planar cell is related to the splay elastic constant by the relation $V_{\text{th}} = \sqrt{\pi^2 K_1 / \varepsilon_0 \Delta\varepsilon}$ (see Supplementary Note 4).

**Normalised density changes**. Changes in density were measured by polarizing optical microscopy. A section of a capillary of 500 µm width and 50 µm was filled with the material and sealed at both ends. The capillary was placed between slightly uncrossed polarizers for better definition of the meniscus position. The sample was heated to the isotropic phase and then cooled at 2 °C/min while recording.

Temperature gradients were avoided with a copper enclosure for the capillary. Changes in density are then calculated from the variation of the total material slab length referenced to the isotropic phase, $\rho(T)/\rho_{iso} = V_{iso}/V(T) = L_{iso}/L(T)$.

## Data availability

The data that support the findings of this study are available from the corresponding author upon reasonable request.

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

## Acknowledgements

Computational work was undertaken on either ARC3 or ARC4, part of the High Performance Computing facilities at the University of Leeds, UK, or the YARCC or Viking High Performance Computing facilities at the University of York. N.S and A.M. acknowledge the financial support from the Slovenian Research Agency (research core Funding No. P1-0192). J.M.-P. acknowledges funding by the University of the Basque Country (project GIU18/146). N.S. and J.M.-P. thank Prof. M.R. de la Fuente for technical support with the dielectric measurements.

## Author contributions

R.J.M. synthesised the materials, performed X-ray experiments, DFT and MD simulations. N.S. and J.M.-P. performed dielectric spectroscopy measurements and contributed to its analysis. N.S. and A.M. carried out DLS experiments. N.S., J.M.-P. and A.M. collected birefringence data. N.S. and A.M. coordinated the work. A.M. oversaw all the contributions. R.J.M., N.S. and A.M. prepared the initial draft of the manuscript and all the authors made contributions to the final version.

## Competing interests

The authors declare no competing interests.
