## [Peer Review File · Nature Communications]

Reviewers' Comments:

Reviewer #1:

Remarks to the Author:

This paper deals with the ferroelectric nematic (NF) phase, which was recently discovered in a few liquid crystal materials. A subtle dependence on chemical details was found, and the molecular mechanism driving the nematic to NF phase transition is an open question. The paper is focused on the comparison between two structurally similar compounds, one of which (RM734) forms the NF phase, whereas the other (RM734-CN) exhibits only a conventional nematic phase. An extensive experimental investigation of the latter is presented, and the results are compared with those previously reported for the former. The experimental investigation is complemented by computational studies, Quantum Chemical (QC) calculations and Molecular Dynamics (MD) simulations, for both compounds. Some distinctive differences between the two systems are identified.

The topic has fundamental and applicative relevance and has high current interest, also beyond the liquid crystal community. The results are original and provide new insight on the difference between the conventional nematic phase formed by systems that do or do not exhibit the NF phase. Possible reasons, at the molecular level, for these differences are suggested; however, the suggestions are not fully substantiated, and this weakens the impact of the study. Some more specific comments are reported below.

All this considered, I cannot recommend publication in Nature Communication of this paper in its present form, but I think that a revised version, where the questions pointed out below are carefully addressed, could be considered for publication.

1) Molecular Dynamics simulations

Since the results of MD simulations are relevant for the conclusions, this part of the study needs special attention.

1a) The quality of the Force Field could be crucial when, as in the present case, subtle effects dependent on chemical details are investigated. Here it is not clear if/how the Force Field was validated.

In particular, for the systems and the effects investigated in this paper, an adequate treatment of electrostatics could be important. One may wonder if polarizability should be included (i.e. a polarizable Force Field), as in Ref. 18.

Here RESP charges were used; were they rescaled? (to my knowledge, RESP charges tend to be overestimated)

1b) Due to the long time required for molecular flipping, samples were frozen in polar/apolar states, so here 'equilibration' is not to be intended in a thermodynamic sense. In my opinion this is an important point, which should be explicitly commented and the physical meaning of the results obtained under such simulation conditions should be commented.

1c) From MD simulations one can obtain information on the positional correlations, which presumably are crucial for the formation of the NF phase; I think that this should be absolutely included in the study (see for instance Ref. 18).

1d) A detailed comparison with Ref. 18, which reports MD simulations for RM734, one the two compounds investigated here, would provide useful insights on the reproducibility of results and the effects of the Force Field.

1e) Fig. 5: the N/I transition seems to be extremely broad (in current simulations of other mesogenic systems ranges of around 10 K are reported), what is the reason?

As further comments on this figure:

- it could be useful to see on the same plot also P1 as a function of temperature;
- since these systems are experimentally found to crystallize at temperatures higher than 400 K, I miss the reason for showing data for nematic phases at temperatures as low as around 250 K.

2) X-ray scattering

This part of the paper, in particular what regards SAXS experiments, is very interesting and in my opinion it should be strengthened.

2a) Details on the calculation of SAXS profiles should be given, maybe in the Supporting Information.

2b) To my knowledge CRY SOL was developed for macromolecules in solution; it should also be explained how it was adapted to the case of thermotropic liquid crystals and how the results were validated (unless there are references in the literature, which in this case should be introduced).

2c) An interpretation of the features of the SAXS profiles for polar systems should be provided, e.g. in relation to the information on the positional correlations obtained from MD trajectories. (See also point 1c.) The following sentence is rather generic and should be substantiated:

"Although such molecular shift seems to be the origin of the additional SAXS peaks, the small positional correlations evidenced by the weak scattering signal suggest that some staggered positions are only slightly more favourable than others."

3) Density

I find that the considerations on density are a little confusing. A decrease of density on decreasing temperature is the usual behavior in phase transitions, and can be ascribed to the combination of more efficient packing and improved attractive interactions in the low temperature phase (I presume that both have to be considered for thermotropic systems). So, the increase of density at the N-NF transition does not seem to me surprising.

The point is to catch the reason behind such a change. I have the impression that this study has the potential to shed light on this, if the relevant information is extracted by a careful analysis of MD trajectories and scattering profiles (see points 1c, 2c and 4a).

4) Quantum Chemical calculations

4a) Difference of a few degrees in the angle between the molecular axes, as obtained from quantum chemical calculations for a pair of molecules, are assumed to be crucial for the formation of the NF phase. I think that this assumption should be better substantiated.

- It is not obvious that the results obtained for a pair of molecules can provide useful insights for the phenomenon under investigation. Actually, geometric details obtained for isolated pairs could correspond to configurations that are scarcely representative in a polar environment. I think that the results obtained by QC calculations for pairs of molecules should be compared with the analogous geometric parameters obtained from MD trajectories; in the paper there is a sentence ("The staggered parallel pairing suggested by DFT calculations is also observed in the bulk phase during molecular dynamics simulations of the polar nematic configuration of RM734 (Fig.SI.9)."), but it is rather generic and should be supported by data.

- Considering the results reported in Table 1, I notice that, whereas the angles between molecular axes are assumed to be so significant, the interaction energies (I suppose that ΔE is the difference between the energy of the pair of molecules and twice the energy of a single molecule) are fully ignored. The latter data indicate a strong energetic preference for the antiparallel orientation for both compounds, which is hard to reconcile with the formation of a polar phase. (By the way, this preference is the opposite of the results reported in Ref. 18.) I miss the reason why one can use independently geometry and energy information

obtained from the same calculations (in other words: if energies calculated for isolated pairs cannot be taken as meaningful in relation to the problem under investigation, why can geometries be assumed to be significant?)

4b) At page 8 definition of the angle between molecules is not immediately clear (by the way, in the text there seems to be some confusion between C12 and C13). I suggest to show in a figure the 'molecular axes' that are used to calculate this angle.

4c) Minor comments

- Since the results of single molecule calculations are not particularly relevant for the present study, the plots of torsional potentials in Fig. 4 could be moved to the Supporting Information. The space in the figure could be better used to enlarge the images of pairs of molecules, which in the present form are too small to distinguish differences.

- A figure showing RESP charges could be introduced in the Supplemental Information.

5) Other comments

- RM734CN forms a nematic phase in a relatively restricted range, between 173°C and 200°C (see Fig. 1). However measurements at temperatures down to 120°C are reported. I think that this should be commented. Also the meaning of the brackets in "(Ns 139.7)" in Fig. 1 should be explained.

- The dielectric relaxation data for RM734-CN show a discontinuity (separation of 2 slow relaxation modes and appearance of a fast mode) at around 170 °C. Is it accidental that this occurs around the Cr-N transition?

- Fig 5c: the lines for the polar and the apolar phase cannot be distinguished; a different representation should be used.

- Caption to Table 1:

Delta E_{int} should be defined (only E_{int} is defined). Also a definition of the 'complexation angle' could be useful (although this is defined in the text).

- Caption to Fig. SI.2: "... squares ... circles" there seems to be a mistake in the definition of symbols.

Reviewer #2:

Remarks to the Author:

This manuscript presents very interesting and important results on physical properties and molecular ordering of RM734 which shows a ferroelectric splay nematic phase and RM734-CN which shows just a classical nematic phase. Observed different behavior between RM734 and RM734-CN could be very useful to gain insight into the driving mechanism for the formation of the nematic polar ordering, that is, a ferroelectric splay nematic phase. Such works are valuable and significant for development of liquid crystal science. On the other hand, it is unfortunate that molecular origin of the polar nematic phase is not fully elucidated in this manuscript.

The authors state that a reduction of excluded volume is the origin of the polar nematic phase. However, I have a doubt about cause and effect of this phenomenon. Isn't it possible to say that the excluded volume is reduced because a polar order of the molecules appears? That is a question of which came first, the chicken or the egg. If a reduction of excluded volume is the origin of the polar nematic phase as the authors state, many other compounds should show polar nematic phase. But actually that is not true. I think that there

could be other more direct causes of the polar nematic ordering.

I wonder if the conclusion drawn in this manuscript is limited to only the two compounds because the study and the comparative investigation are mainly made for the two compounds, RM734 and RM734-CN only. I don't believe that the results obtained in this study can contribute to universal understanding of the molecular origin of the polar nematic phase.

The authors performed molecular dynamics simulations to confirm whether polar (parallel) or apolar (antiparallel) ordering is dominant for RM734 and RM734-CN. The production MD run of each simulation was 250 ns from both polar and apolar starting configurations. This running time is too short to identify the trend. As shown in dielectric relaxation spectra in Figure 2, the relaxation frequency of collective reorientation of the dipole moments ranges around 10k Hz or less at 130 degree Celsius, that is, the relaxation time is 10k ns or more, which is much longer than MD simulation run time performed in this study.

Reviewer #3:

Remarks to the Author:

Accept, subject to corrections.

The discovery of the N_s phase by the authors is one of the most important made in recent few years in the subject of liquid crystals. Following the discovery of the Twist-Bend Nematic phase at the start of the decade, interest in alternative nematic phases has been intense; arguably the most important discovery that could be made in this field is that of a ferroelectric nematic, which it what is reported by the current authors work. The authors have made a number of prior publications on this topic, and so it is important to validate whether or not this paper is of sufficient new importance in its own right, to make it suitable for Nature Communications.

The current paper seeks to do a very detailed study of two similar materials: the first found N_s material RM734, and an equivalent molecule with a terminal CN rather than terminal NO₂ (ie RM734CN). It compares the various relevant physical properties (the measurements for the RM734CN are all new) and importantly an in-depth comparison of the Molecular Dynamics for the two structures. I believe that this level of detail is exactly what is required and is a most important question. This makes the subject of the paper and the presented results certainly worthy of publication in Nature Communications.

I have some reservations on whether or not the text as it stands does sufficient justice to the results, and believe that the authors should consider the following general comments:

1: The behaviour of RM734CN is far more unusual than the authors give credit. That is, unlike a conventional nematic, it shows a strong collective dipole mode (though weaker than the N_s material it is still exceptionally strong compared to conventional nematics); it shows the same unusually low splay elasticity (although it does not show the critical decrease that occurs immediately above the N_s transition which it does not exhibit in the temperature range that is accessible through the hysteresis before freezing to the crystal form.

2: Given the importance of the results, I believe that the paper would be greatly improved by including data for RM551 (a second N_s material included right at the end of the paper) and a compound that is still further removed from the polar behaviour than RM734CN (I imagine RM734CN without the Methoxy side group should be sufficiently non-wedge shaped). That is, there needs to be validation of the N_d behaviour throughout the paper, rather than just at the end. There also needs to be clarification that the unusual behaviour observed by the RM734CN material (even though it does not show an N_s phase across the accessible temperature range) is at the borderline to N_s formation. That is, to show that the highly polar nature of RM734CN does not occur when its wedge shape is reduced (through removal of the transverse methoxy, or much longer terminal alkoxy group but without the fluorination used in RM551, etc) rather than the changes associated with the terminal dipole alone.

3. It would be harsh to condemn the paper to a lesser journal if the authors do not already have the results that show more conventional behaviour from a larger molecular deviation than RM734CN alone. The authors have correctly found a very close analogue in that material, and therefore how sensitive the formation of the phase is. The fact that the phase is also shown in RM551 (as are the differences recorded in table 1) is . However, if they cannot add such results from a molecular variant that is FURTHER from the formation of the Ns phase than RM734CN they should at least include some comments on this in the discussion.

4. Although it is not clear from the text, the results are largely from nematic and Ns phases supercooled below their melting temperatures. It is not clear from the description for the reader, whether or not the MD simulations suggest that RM734CN would never form the Ns phase even if supercooled to a much lower temperature than is possible practically due to crystallisation, or whether it would eventually form the Ns phase at some much lower temperature. This latter possibility seems precluded by the fact that the the Ns phase is quenched by just a 10% mixture of the RM734CN, but it is not clear why this effect would be so strong from the MD.

5. The inclusion of RM551 was very powerful. It should be said at the outset that a model will be presented for the formation of the NS phase that is then tested successfully against a second material. (As I state above, I would like to also see a second negative prediction too, for something less polar than the RM734CN, though that may not be possible).

Below are comments written during the first reading of the text that may be apposite. Some of these may already have been covered above.

The introduction states that it is important to understand the molecular design rules, and the paper concentrates on swapping the terminal nitro for a terminal cyano moiety. However, the materials both use methoxy terminated mesogens, which inherently will lead to higher crystallisation temperatures. At the end of the paper, the authors include a longer terminal alkyl group from their prior art, and found the methoxy also essential for the formation of the Ns phase. However, some comment on this would be helpful for the reader in the introductory remarks.

Figure 2. The caption should make clear that RM734CN was used for a). It would also be useful to include similar results to these for RM734. These are found by going to the SI of reference [3]. Repeating in the present SI would be beneficial.

The results shown here for RM734CN are ambiguous. This is because the crystallisation point for this material is quoted as 173°C with no mention of hysteresis on heating and cooling. The text describing the diagram suggests that the 1-mode relaxation is Arrhenius across the nematic phase, but no results are shown on the figure for this: only for the 2-mode. There are results for all 3 relaxations 1,2,3 from 120°C to 172°C. Are these from a supercooled nematic phase, or measured in the crystal phase? Alternatively, the statement in the text that $m_{||,1}$ relaxation frequency of RM734CN is Arrhenius in its nematic phase might better apply to $m_{||,2}$; this is the only relaxation shown in the range 172°C - 200°C that corresponds to the nematic temperature range. I presume that this lack of clarity could be corrected by a simple statement earlier that the samples supercooled in the nematic phase to below 120°C. The minimum temperature for obtaining the Ns phase in RM734 would also be instructive, though the results shown in Fig 2 seem to only go to 130°C, which is only 2-3°C into the Ns phase.

The dielectric results of RM734CN in Fig 2 show a polar, collective relaxation $m_{||,1}$ in the crystal (or supercooled N) phase. It is stated that this is weak, but it is surprisingly strong - with a dielectric strength of 10. This is very large indeed if compared to say similar nematic phases (eg 5CB) that shows no collective motion. It is about half the magnitude of the higher frequency dielectric relaxation of the longitudinal dipole moment $m_{||,2}$. Given this, it would be apposite to make some comment in this section that it is still high even though no Ns phase was found for this material, rather than just stating that it is much weaker than for the compound RM734 that does show the Ns phase. That is, arguably, it is the similarity between the materials (low K11, pre-transitional collective dielectric mode) that unusual given that the material does not form an Ns phase.

For RM734, only the pretransitional behaviour for the Ns phase is shown, and there are no results presented for below the N-Ns transition. Some comment on why this is should be included here (presumably it appeared in ref [3]). Presumably (given fig 3) the relaxation becomes very low frequency as the viscosity diverges.

Again on the subject of how SIMILAR RM734 and RM734CN are, both have unusually low K11. The difference is that there is a critical pretransitional effect in RM734 above Ns-N but this occurs over a narrow temperature range. One can envisage that RM734CN might super cool to a sufficiently low temperature and then form the Ns phase and that critical behaviour then be observed. That is, it is not clear from the paper whether or not RM734CN could exhibit the Ns phase at some temperature, or whether its configuration prevents this from ever happening. The discussion begins with discussion that seem to preclude this possibility, but it is not clear how that conclusion could be made.

Page 7 - dipole vectors need units.

SI

Spelling of "consequently".

Fig S1-2 points are mislabelled s squares and circles rather than open and closed points.

Thank you for the prompt review of this manuscript. We have considered the referees point in some detail, and have performed extensive new simulation studies of new materials, and we offer this response.

NATURE COMMS REVIEW

REVIEWER COMMENTS

Reviewer #1 (Remarks to the Author):

This paper deals with the ferroelectric nematic (NF) phase, which was recently discovered in a few liquid crystal materials. A subtle dependence on chemical details was found, and the molecular mechanism driving the nematic to NF phase transition is an open question. The paper is focused on the comparison between two structurally similar compounds, one of which (RM734) forms the NF phase, whereas the other (RM734-CN) exhibits only a conventional nematic phase. An extensive experimental investigation of the latter is presented, and the results are compared with those previously reported for the former. The experimental investigation is complemented by computational studies, Quantum Chemical (QC) calculations and Molecular Dynamics (MD) simulations, for both compounds. Some distinctive differences between the two systems are identified.

The topic has fundamental and applicative relevance and has high current interest, also beyond the liquid crystal community. The results are original and provide new insight on the difference between the conventional nematic phase formed by systems that do or do not exhibit the NF phase. Possible reasons, at the molecular level, for these differences are suggested; however, the suggestions are not fully substantiated, and this weakens the impact of the study. Some more specific comments are reported below.

All this considered, I cannot recommend publication in Nature Communication of this paper in its present form, but I think that a revised version, where the questions pointed out below are carefully addressed, could be considered for publication.

1) Molecular Dynamics simulations

Since the results of MD simulations are relevant for the conclusions, this part of the study needs special attention.

1a) The quality of the Force Field could be crucial when, as in the present case, subtle effects dependent on chemical details are investigated. Here it is not clear if/how the Force Field was validated.

We agree that comment on this is necessary. The modified general amber force field for liquid crystals (GAFF-LCFF) which we employ, has been validated previously and gives significant improvements over conventional GAFF in terms of prediction of heats of formation and densities, but also in terms of its ability to generate thermodynamically stable liquid crystalline order. [10.1039/C5CP03702F]

We had added text to this effect in the materials and methods section:

We used the General Amber Force Field (GAFF)⁴² with modifications for liquid crystalline molecules (GAFF-LCFF)⁴³ as this gives significant improvements over conventional GAFF in terms of predicting heats of formation and density, but also in term of its ability to generate thermodynamically stable liquid crystalline order.

In particular, for the systems and the effects investigated in this paper, an adequate treatment of electrostatics

could be important. One may wonder if polarizability should be included (i.e. a polarizable Force Field), as in Ref. 18.

Combining polarizability with GAFF-LCFF is non-trivial and would be a major undertaking, but this is certainly an interesting suggestion, and one that we will look to address in future. We note that, in Ref 18, there is also a comment that both polarizable and non-polarizable forcefields were explored.

Here RESP charges were used; were they rescaled? (to my knowledge, RESP charges tend to be overestimated)

RESP charges were not rescaled.

1b) Due to the long time required for molecular flipping, samples were frozen in polar/apolar states, so here 'equilibration' is not to be intended in a thermodynamic sense. In my opinion this is an important point, which should be explicitly commented and the physical meaning of the results obtained under such simulation conditions should be commented.

We agree with the referee that this is an important point, which requires additional clarification from us. We obtain polar/apolar states by rapidly compressing a low density simulation comprised of molecules aligned parallel or 50:50 antiparallel to a liquid-like density of $\sim 1 \text{ g cm}^3$. This rapid compression preserves much of the parallel or antiparallel order, which is allowed to evolve throughout the subsequent production MD run of 250 ns, with no constraints applied to enforce the polar or apolar states. Nevertheless head-tail flipping does not occur to an appreciable extent, as evidenced by the stable values of P1 a function of time (i.e. the polar order parameter, $P1 = \cos\langle\theta\rangle$) over the course of the MD simulation.

We have clarified the portion of the manuscript that details our simulation setup to read:

<< ORIGINAL TEXT >>

We constructed initial low density lattices of 680 molecules with random positional order. For polar nematic simulations, all molecular dipole moments were oriented along +x, while apolar nematic simulations have a 1:1 mixture of molecules aligned along +x and -x. Following energy minimisation, the simulation box was then compressed over 50 ps to a mass density of $\sim 1 \text{ g cm}^3$ which is typical of that of low molecular weight liquid crystals. Following compression, simulations were allowed to equilibrate for 30 ns, with subsequent production MD runs of a further 250 ns.

<< NEW TEXT >>

We constructed initial low density lattices of 680 molecules with random positional order. For polar nematic simulations, all molecular dipole moments were oriented along +x, while apolar nematic simulations have a 1:1 mixture of molecules aligned along +x and -x. Following energy minimisation, the simulation box was then compressed over 50 ps to a mass density of $\sim 1 \text{ g cm}^3$ which is typical of that of low molecular weight liquid crystals. Following compression, simulations were allowed to equilibrate for 30 ns, with subsequent production MD runs of a further 250 ns. No constraint were applied to enforce or preserve dipole order, which was allowed to evolve throughout the production MD run. Head-to-tail flipping of molecules is rarely seen in the nematic phase, as evidenced by the stable values of $P1 = \cos(\theta)$, where θ is the angle between a given molecular inertia axis and the nematic director (Supplementary Figure 21).

We have also added further information in the SI:

<< ADDITION TO SUPPLEMENTARY INFORMATION >>>

Supplementary Figure 21. Plot of the $\langle P_1 \rangle$ order parameter as a function of simulation time for *RM734* in the polar and apolar configurations at 400 K, and its mean value in both.

As shown in Supplementary Fig. 21, the stable value of $\langle P_1 \rangle$ indicates that head-tail flipping of molecules is largely arrested; however, as shown in the manuscript, when nematic simulations are heated into the isotropic liquid $\langle P_1 \rangle$ becomes near-zero, demonstrating that the lack of flipping is a phenomenon associated with nematic order rather than the specific molecules studied herein.

1c) From MD simulations one can obtain information on the positional correlations, which presumably are crucial for the formation of the NF phase; I think that this should be absolutely included in the study (see for instance Ref. 18).

This is a very good suggestion and has enormously strengthened the manuscript. Typically, one analyses the positional correlations of an MD simulation by counting the number of particles within a spherical shell of some position, giving an (isotropic) radial distribution function. The analysis we have performed in this revision, and we understand this is the same analysis used in Ref 18, is to use a cylindrical shell oriented with its length along the director – this allows us to obtain the anisotropic positional correlations shown below. As with Ref. 18, this reveals differences between the polar and apolar states. Going beyond Ref. 18, we are able to demonstrate differences between *RM734* and *RM734-CN* that offer an explanation as to the molecular origins of this phase.

<< NEW TEXT >>

We calculate pair-correlation functions for *RM734* and *RM734-CN* in both polar and apolar nematic states from the centre-of-mass of each molecule via a cylindrical binning procedure (Supplementary Figure 13). Peaks indicate preferred pairing, and in each analysis steric repulsion between molecules leads to a region of low probability centred at $h, r=0$ (Figure 5e-h). *RM734* in its polar configuration (Figure 5e) displays large on-axis arcs at $h \approx \pm 20$, which corresponds to head-to-tail pairing (e.g. terminal OMe to terminal NO₂); this feature is somewhat less pronounced in *RM734-CN* in the polar configuration (Figure 5g), indicating a reduced preference for these configurations. The polar configuration of *RM734* also exhibits several off-axis peaks which are absent in the apolar configuration and in *RM734-CN* in both polar and apolar configurations. In the polar configuration, *RM734-CN* has significantly stronger off-axis peaks at $h \approx r \approx 6$ Å than *RM734*; these corresponding to staggered parallel pairs of side-by-side molecules.

There is a subtler difference between the different packing modes in the apolar simulations (Figure 5f and Figure 5h), with a mixture of head-to-head and side-side pairs being common to both materials. The on-axis peaks at $h \approx \pm 20$, for both materials, correspond to head-to-head (or tail-to-tail) configurations. The off-axis peaks at $h = r = 6$ Å correspond to staggered antiparallel pairs, which are known to be dominant for other low-molecular weight nitrile terminated materials (e.g. 5CB).

Our analysis of *RM734* is in accord with the earlier simulation study by Chen et al.¹⁸, and demonstrates that this material exhibits a strong preference for head-to-tail pairing. By also simulating *RM734-CN*, we find that the terminal nitrile is seemingly unable to sustain this pairing motif, instead adopting a staggered parallel pairing, driven by CN-C(O)O interactions, in polar

simulations. We suggest that the head-to-tail pairing is presumably crucial to the formation of the polar N_S phase under experimental conditions.

<< NEW SUPPLEMENTARY FIGURE >>

Supplementary Figure 13: An instantaneous configuration of RM734 in the polar nematic configuration with a single molecule shown as space-filling model, and the molecular centre of mass shown for all remaining molecules; (b) Illustration of the cylindrical averaging process for a single molecule oriented with its mass inertia axis along X.

<<AND SEE NEW FIGURE 5 BELOW – Page 5 of this response document>>

1d) A detailed comparison with Ref. 18, which reports MD simulations for RM734, one the two compounds investigated here, would provide useful insights on the reproducibility of results and the effects of the Force Field.

We are in agreement that a comparison adds insight and strengthens the manuscript. We have added the following text on page 12:

<<NEW TEXT>>

Briefly, comparing our MD simulations of RM734 with those reported in Ref 18, we find our simulations give near identical values of P2 (polar = 0.787, apolar = 0.78, both at 130 °C in ref 18). Values of the polar order parameter, P1 are also comparable (polar = 0.924, apolar = 0.013, both at 130 °C) although our values are slightly lower for the polar simulation. In Ref. 18 the simulated mass density is reported as 1.3 g cm³ “at 130 °C”, but no distinction is made between the polar and apolar nematic states; this value is broadly comparable with our own simulations of RM734. Important points of differentiation between this prior MD work and our own are the number of molecules per simulation (368 in ref 18 versus 680 here), production MD simulation length (“... in excess of 20 ns” in ref 18 versus ~250 ns here), and forcefield choice (polarizable APPLE&P in ref 18, non-polarizable GAFF-LCFF here).

<<END OF NEW TEXT>>

We note that despite the differences between the setup of the two simulations the force fields used and their size (both number of molecules and simulation time), the bulk properties are largely consistent between our work and the earlier simulation work in Ref. 18.

1e) Fig. 5: the N/I transition seems to be extremely broad (in current simulations of other mesogenic systems ranges of around 10 K are reported), what is the reason?

It is not that the transition is broad, but rather we performed MD simulations in discrete temperature increments of 50 K. We think this may be a result of the plotting style used in Figure 5b; we used solid lines, whereas the actual $\langle P_2 \rangle$ values occur at discrete temperature intervals. We have changed the plotting styles to markers and lines, and added a note in the figure legend to say that lines are a guide to the eye (revised figure given below).

As further comments on this figure:

- it could be useful to see on the same plot also P_1 as a function of temperature;
- since these systems are experimentally found to crystallize at temperatures higher than 400 K, I miss the reason for showing data for nematic phases at temperatures as low as around 250 K.

Showing P_1 as a function of temperature is a smart suggestion; a plot has been included in the revised Figure 5 (see below).

On temperature, we would add that while these materials have a melting point of ~ 400 K, they do not immediately solidify when cooling below 400 K. Under certain experimental conditions, it is possible to cool RM734 and RM734-CN to ambient temperature (i.e. 300 K, see new Supplementary Fig. 2 for RM734) without crystallisation, and so we consider that including simulation data at these temperatures is valid given that they are experimentally accessible. After reflecting on the referees comment, we agree that simulations at 250 K were probably excessive; however given that they are completed and do not affect the conclusions of the manuscript we are in favour of retaining them if the referee does not object. We would add that while we do not have data to say it is impossible to access a temperature of 250 K experimentally, experience would suggest the sample is likely to solidify within a few minutes at this temperature.

Fig. 1: Molecular dynamics simulations. (a) Snapshots of molecular dynamics simulations of RM734 and RM734-CN in polar and apolar configurations, and at temperatures of 400 K and 500 K, respectively. In each plot, the coloured lines represent the molecular long axis. The order parameter of each molecule is represented by the colour depth of its corresponding line representation. Lastly, two colour scales are used: black to blue, where the molecular long axis makes an angle of $\leq 90^\circ$ with the director, and black to red when this angle

is $\geq 90^\circ$ with the director. Each snapshot was taken at $t = 250$ ns. Plots of the order parameters P1 (b) and P2 (c) as a function of simulation temperature for RM734 and RM734-CN in polar and apolar configurations. Plots of the difference in mean density between the polar and apolar simulations of RM734 and RM734-CN as a function of temperature (d). In (b-d) data points represent mean values from simulations at the indicated temperature, whereas solid lines are a guide to the eye. (e-h) cylindrical pair correlation functions obtained for MD simulations of RM734 (e, f) and RM734-CN (g, h) in polar (e, g) and apolar (f, h) nematic configurations, respectively, at 400 K. The cylindrical pair correlation functions were computed over the same time window as was used in the calculation of X-ray scattering data, i.e. 200-280 ns.

2) X-ray scattering

This part of the paper, in particular what regards SAXS experiments, is very interesting and in my opinion it should be strengthened.

2a) Details on the calculation of SAXS profiles should be given, maybe in the Supporting Information.

We calculate SAXS profiles *via* two methods: 1D data from CRY SOL, and 2D data according to B. Coscia et al. From the original papers, which we cite, readers are able to access these software packages.

In our original submission it was mentioned, in passing, that we use a “modification” to the method of Coscia et al; explicitly, the software was altered to run under a Windows software environment (as opposed to Unix) and also to save the calculated 2D SAXS patterns as a numerical array (rather than an image only as provided in the unmodified software). These modifications were made so that data could be then interfaced with existing software tools for processing of experimental X-ray data, without the limitations imposed and are trivial to make. The actual calculation engine, which is available *via* GitHub and linked to in the original paper by Coscia *et al*, of the software is unmodified.

2b) To my knowledge CRY SOL was developed for macromolecules in solution; it should also be explained how it was adapted to the case of thermotropic liquid crystals and how the results were validated (unless there are references in the literature, which in this case should be introduced).

The referee is correct that CRY SOL was developed for macromolecules in solution. CRY SOL is, however, also capable of calculating scattering profiles for isolated molecules, without a hydration/solvent layer, which is how we utilise it. By using a given MD trajectory frame as an input (as opposed to an isolated molecule), we use CRY SOL to calculate X-ray scattering intensity of said frame. These are then averaged, to afford the data presented in the manuscript.

The advantage of using CRY SOL is that it offers the user significant control over the resolution and Q range of the resulting calculated scattering profile – other tools (e.g. *GMX SAXS* in GROMAXS, *compute xrd* in LAMMPS) are less useful to us in this respect. Where CRY SOL is less useful is that it produces a 1D curve rather than a 2D SAXS pattern, and also the scattering profile is an average of all orientations (i.e. anisotropy is neglected). However, this is sufficient for comparing scattering vectors of experiment and simulation. No adaption to CRY SOL was required for use with thermotropic liquid crystals; results were validated by comparing experimental SAXS profiles with those calculated from MD simulations, which is presented in the manuscript.

2c) An interpretation of the features of the SAXS profiles for polar systems should be provided, e.g. in relation to the information on the positional correlations obtained from MD trajectories. (See also point 1c.) The following sentence is rather generic and should be substantiated:

“Although such molecular shift seems to be the origin of the additional SAXS peaks, the small positional correlations evidenced by the weak scattering signal suggest that some staggered positions are only slightly more favourable than others.”

We feel that the addition of the pair correlation functions obtained from MD simulations and their analysis (Fig 5e-h and Supplementary Figures 14-15) allows us to substantiate this statement. Due to the new additions of the text, we rephrased this statement:

<<ORIGINAL TEXT>>

The staggered parallel pairing suggested by DFT calculations is also observed in the bulk phase during MD simulations of the polar nematic configuration of RM734 (Fig.SI:9). Although such molecular shift seems to be the origin of the additional SAXS peaks, the small positional correlations evidenced by the weak scattering signal suggest that some staggered positions are only slightly more favourable than others.

<<NEW TEXT>>

The positional correlations are small, evidenced by the weak scattering signal, suggesting that some staggered positions are only slightly more favourable than others, which is further evidenced by the pair correlation plots (Figure 5e-h) and the radial distribution functions (RDFs) (Supplementary Figure 14-15).

3) Density

I find that the considerations on density are a little confusing. A decrease of density on decreasing temperature is the usual behavior in phase transitions, and can be ascribed to the combination of more efficient packing and improved attractive interactions in the low temperature phase (I presume that both have to be considered for thermotropic systems). So, the increase of density at the N-NF transition does not seem to me surprising.

The point is to catch the reason behind such a change. I have the impression that this study has the potential to shed light on this, if the relevant information is extracted by a careful analysis of MD trajectories and scattering profiles (see points 1c, 2c and 4a).

We agree with the referee that it is not so much the behaviour of density that is of interest, but the underlying reason and – as we wrote in the title of the manuscript – its molecular origins.

The main difference between the RM734 and RM734-CN is that, at the same temperature, for the first the polar phase is denser than the apolar, while for the second this is not the case. This, we believe, is the mechanism of the growth of polar correlations in RM734 already at higher temperatures in the nematic phase. While apolar phase is entropically more favourable than the polar phase, when the polar phase is higher in density than the equivalent apolar configuration it is favoured from an excluded volume point of view. As evidenced by the dielectric spectroscopy, polar correlations in RM734-CN also exists but they do not grow because polar and apolar phase have the same density and there is no gain in excluded volume in polar with the respect to apolar phase which would drive the growth of the polar correlations.

We also feel that our additional simulations on structurally related materials, requested by referees 2 and 3, add further weight to this argument. For the sake of brevity, and to avoid repetition in our response, we will describe this only briefly here (further comment is on page 15 of this response).

Our additional MD simulations reveal that the significant increase in density seen for the polar nematic configuration (versus the apolar nematic) is only present for *RM734* and *RM554*; when we replace the NO_2 with a $-\text{CN}$ (*RM734CN*), remove the lateral OMe group (*RM63*), or extend the terminal chain (*RM500*), then the disparity in density is absent. Experimentally, the N_s phase is only observed for the two materials which display a large difference in density between polar and apolar nematic configurations, and is absent for the other 3.

4) Quantum Chemical calculations

4a) Difference of a few degrees in the angle between the molecular axes, as obtained from quantum chemical calculations for a pair of molecules, are assumed to be crucial for the formation of the NF phase. I think that this assumption should be better substantiated.

I hope the referee can appreciate our intent with this section of the original submission, but with the addition of pair-correlation functions extracted from the MD simulations this part of the manuscript is now redundant. We feel it is important to retain work on calculated molecular parameters (e.g. dipole moment, polarizability, torsion plots, charge distribution) as one might expect these to correlate with the incidence of the polar N_s phase (however they do not). The inclusion of pair-correlation function analysis, suggested by the referee, renders the DFT study of intermolecular interaction energies obsolete, and so it has been removed because of other more important additions.

*- It is not obvious that the results obtained for a pair of molecules can provide useful insights for the phenomenon under investigation. Actually, geometric details obtained for isolated pairs could correspond to configurations that are scarcely representative in a polar environment. I think that the results obtained by QC calculations for pairs of molecules should be compared with the analogous geometric parameters obtained from MD trajectories; in the paper there is a sentence ("The staggered parallel pairing suggested by DFT calculations is also observed in the bulk phase during molecular dynamics simulations of the polar nematic configuration of *RM734* (Fig.SI.9)."), but it is rather generic and should be supported by data.*

We are in agreement with the referee; in fact, our analysis of MD trajectories supports the referees suggestion that these isolated pairs may be scarcely populated in the polar environment, hence we have removed this section of the manuscript.

- Considering the results reported in Table 1, I notice that, whereas the angles between molecular axes are assumed to be so significant, the interaction energies (I suppose that ΔE is the difference between the energy of the pair of molecules and twice the energy of a single molecule) are fully ignored. The latter data indicate a strong energetic preference for the antiparallel orientation for both compounds, which is hard to reconcile with the formation of a polar phase. (By the way, this preference is the opposite of the results reported in Ref. 18.) I miss the reason why one can use independently geometry and energy information obtained from the same calculations (in other words: if energies calculated for isolated pairs cannot be taken as meaningful in relation to the problem under investigation, why can geometries be assumed to be significant?)

ΔE is simply the difference between the sum of monomers and the counterpoise corrected energy of the n -mer complex; although this has now been removed.

We agree with the referee however. That there is no indication that these are the lowest energy complexes that could be formed; consider that there are many different orientations and positions we could explore, and to do so would require hundreds if not thousands or even tens of thousands of

different starting points to be considered. But this has been removed from the manuscript now, as the analysis of MD trajectories, completed at the referees suggestion, accomplishes the same aim.

4b) At page 8 definition of the angle between molecules is not immediately clear (by the way, in the text there seems to be some confusion between C12 and C13). I suggest to show in a figure the 'molecular axes' that are used to calculate this angle.

We would have corrected C₁₂/C₆₇ in the text to C₁₃/C₆₇ – thank you for pointing out this typo. We feel the description of which atoms this refers to (the distal aromatic carbon atoms) is sufficient to convey our intended meaning. But, as discussed, this section has been removed.

4c) Minor comments

- Since the results of single molecule calculations are not particularly relevant for the present study, the plots of torsional potentials in Fig. 4 could be moved to the Supporting Information. The space in the figure could be better used to enlarge the images of pairs of molecules, which in the present form are too small to distinguish differences.

We feel that the torsional potentials are important in so much as they demonstrate the similarity between the two materials (on a molecular level), and serve to demonstrate that the occurrence of the Ns phase is not simply explained by considering properties of isolated molecules.

- A figure showing RESP charges could be introduced in the Supplemental Information.

We agree with this good suggestion, and a figure has been included in the Supplemental Information, along with example inputs to Gaussian G09 to generate the RESP charges. Below is the text we have added:

<<NEW TEXT IN SUPPLEMENTARY INFORMATION>>

Supplementary Note 14

RESP charges were calculated at the B3LYP/6-31G(d,p) level of DFT using the Gaussian G09.d01 software package. First, we optimised geometry at the same level of DFT:

```
#p opt b3lyp/6-31G(d,p) nosymm iop(6/7=3) gfinput
```

The RESP charges were then calculated for the optimised geometry by first computing the ESP charge in Gaussian G09.d01:

```
#p b3lyp/6-31G(d,p) nosymm iop(6/33=2) pop(chelpg,regular)
```

And then processing the output with the antechamber program in AmberTools 16.

We give charges for RM734 and RM734CN in Supplementary Figure 20, below.

Supplementary Figure 20: RESP charges calculated at the B3LYP/6-31G(d,p) level of DFT for (a) *RM734* and (b) *RM734CN*.

5) Other comments

- *RM734CN* forms a nematic phase in a relatively restricted range, between 173°C and 200°C (see Fig. 1). However measurements at temperatures down to 120°C are reported. I think that this should be commented. Also the meaning of the brackets in "(Ns 139.7)" in Fig. 1 should be explained.

Having seen the different comments of the referees we agree that the phase sequence of both reported materials as expressed could have been equivocal. Thus, we extended the phase behaviour description for both materials in Materials and Methods, and additionally complemented it with Supplementary Fig. 1 and Supplementary Fig. 2. Detailed description of the added text and Figures can be found below in the response to Referee 3.

- The dielectric relaxation data for *RM734-CN* show a discontinuity (separation of 2 slow relaxation modes and appearance of a fast mode) at around 170 °C. Is it accidental that this occurs around the Cr-N transition?

The discontinuity shown in Fig 2b/c is solely due to the impossibility of uniquely deconvoluting the $m_{||,1}$ and $m_{||,2}$ modes above that temperature during the data-fitting process. The modes overlap in the frequency spectrum and resolving them in a robust way becomes only feasible below that temperature. The same reason applies to the *RM734* data, which shows the ‘splitting’ in two modes below 160 °C. As the reviewer correctly guessed, the fact that it is close to the N-Cr transition in the *RM734-CN* is purely accidental.

We have tried to clarify this in the text by modifying the explanation in the 3rd paragraph of the Results section and extending the description of the fitting procedure in the Supplementary Note 2:

<<ORIGINAL TEXT>>

On cooling, immediately after the *I-N* transition, dielectric spectra is characterized by a relaxation mode at lower frequencies and with growing amplitude. This mode, although close to a Debye at high temperatures, slowly broadens on decreasing the temperature. This same behaviour was observed for *RM734*. As in the latter case, far below the transition, it becomes evident that the single relaxation mode detected in the high temperature range is indeed composed of two relaxation modes, $m_{||,1}$ and $m_{||,2}$.

<<NEW TEXT>>

On cooling, immediately after the I-N transition, dielectric spectra is characterized by a relaxation mode at lower frequencies and with growing amplitude. This mode, although close to a Debye at high temperatures, slowly broadens on decreasing the temperature. This same behaviour was observed for RM734. As in the latter case, far below the transition, **it becomes evident that the seemingly single relaxation mode detected at high temperature range must in fact be deconvoluted into two different relaxation modes**, $m_{||,1}$ and $m_{||,2}$.

- Fig 5c: the lines for the polar and the apolar phase cannot be distinguished; a different representation should be used.

We have made changes to Fig 5; see response to point 1e for the image. Plotting density is rather difficult, because the absolute values are so large with relatively small difference between them. Instead, we plot the difference in density between the two states as a percentage; this gives values which are directly comparable for different configurations of the same material and also the same configuration of different materials.

- Caption to Table 1:

Delta E_{int} should be defined (only E_{int} is defined). Also a definition of the 'complexation angle' could be useful (although this is defined in the text).

There was a missing delta symbol in the legend of Table 1 (E_{int} should be ΔE_{int}); we are grateful to the referee for spotting this error.

We agree that the definition of 'complexation angle' should have been included in the legend for Table 1. However, we note that this section of the manuscript has been removed, as discussed in our response to points above.

Caption to Fig. SI.2: "... squares ... circles" there seems to be a mistake in the definition of symbols.

We thank the referee for spotting this error. Caption has been corrected. "Frequency dependence of the real (full circles) and imaginary (open circles) dielectric permittivity."

Reviewer #2 (Remarks to the Author):

This manuscript presents very interesting and important results on physical properties and molecular ordering of RM734 which shows a ferroelectric splay nematic phase and RM734-CN which shows just a classical nematic phase. Observed different behavior between RM734 and RM734-CN could be very useful to gain insight into the driving mechanism for the formation of the nematic polar ordering, that is, a ferroelectric splay nematic phase. Such works are valuable and significant for development of liquid crystal science. On the other hand, it is unfortunate that molecular origin of the polar nematic phase is not fully elucidated in this manuscript.

We hope the referee will agree that, while the molecular origins are not fully elucidated (similarly they are not entirely elucidated for 5CB, 60 years after its first synthesis) this work presents a large step in

our understanding, only strengthened by the inclusion of several new simulations and analysis of pair-correlations, which reveal different preferred pairing modes for nitro- and cyano- terminated materials to be the probable cause of the Ns phase.

The authors state that a reduction of excluded volume is the origin of the polar nematic phase. However, I have a doubt about cause and effect of this phenomenon. Isn't it possible to say that the excluded volume is reduced because a polar order of the molecules appears? That is a question of which came first, the chicken or the egg. If a reduction of excluded volume is the origin of the polar nematic phase as the authors state, many other compounds should show polar nematic phase. But actually that is not true. I think that there could be other more direct causes of the polar nematic ordering.

The chicken and egg questions are always difficult to address. The formation of nematic phase is driven by a combination of excluded volume contribution and interactions between the molecules. If polar order was mainly a consequence of polar interactions (other than the excluded volume), specific strong correlations between the molecules would be expected, however, the MD simulations and SAXS experiments show that the correlations in the polar phase are weak. Moreover, our results show that in those systems in which the polar phase is experimentally observed, the density in the MD simulation is larger in polar than in apolar configuration, while in those that do not exhibit polar phase the density is the same. In this version of the manuscript, more examples confirming this observation are added.

Entropically, the apolar nematic phase is more favourable than the polar. We argue that, if in polar phase molecules can pack better, i.e., have smaller excluded volume and gain more in translational entropy than they lose in orientational entropy, the polar phase becomes more favourable. To further test this argument, we calculated translational diffusion of molecules for polar and apolar phases of RM734 and RM734-CN. It is expected that the gain in translational entropy is reflected in the increase of translational diffusion constant, which is exactly what is observed. In the polar phase of RM734, the diffusion constant is larger than in the apolar, while for RM734-CN the situation is reversed. If the (attractive) interaction between the molecules was a dominant cause of the polar phase formation, the translation diffusion constant would be smaller in polar phase.

We have added the diffusion constants as extracted from MD simulation in Supplementary Table 3 and revised a sentence in the main manuscript:

<<NEW TEXT in bold >>

Conversely, we find that, at a given temperature, the density of RM734 is somewhat (~ 0.5 %) larger in the polar nematic than in the apolar **and simultaneously molecules exhibit larger translational diffusion constant in polar than apolar state (Supplementary Table 3).**

I wonder if the conclusion drawn in this manuscript is limited to only the two compounds because the study and the comparative investigation are mainly made for the two compounds, RM734 and RM734-CN only. I don't believe that the results obtained in this study can contribute to universal understanding of the molecular origin of the polar nematic phase.

We accept that our original submission was limited by the choice of materials, although we feel the comparison between RM734/RM734CN is quite powerful given their similarity and the fact that they have been characterised by a number of techniques. We hope that the inclusion of additional simulations helps to strengthen our argument (see response to referee 3, page 15). In particular, these additional simulations also show the same correlation between enhanced simulation packing density and the emergence of experimental polar order. Furthermore, the additional analysis we present in this

revision adds further weight to our arguments, and demonstrates an emerging *in silico* predictive capability for designing N_s materials.

The authors performed molecular dynamics simulations to confirm whether polar (parallel) or apolar (antiparallel) ordering is dominant for RM734 and RM734-CN. The production MD run of each simulation was 250 ns from both polar and apolar starting configurations. This running time is too short to identify the trend. As shown in dielectric relaxation spectra in Figure 2, the relaxation frequency of collective reorientation of the dipole moments ranges around 10k Hz or less at 130 degree Celsius, that is, the relaxation time is 10k ns or more, which is much longer than MD simulation run time performed in this study.

Our motivation with MD simulations is not to reproduce the collective reorientation of dipoles observed in dielectric studies, as atomistic MD simulations on 10 μs timescales are not accessible to us currently. We contest that 250 ns is insufficient; it is enough to enable a stable nematic order to emerge in both polar and apolar configurations, which enables us to study simulation density, orientational and polar order, and to simulate X-ray scattering patterns. These are all critically important, as we show (or have shown) that they exhibit behaviour distinct from classical nematics (e.g. high $\langle P2 \rangle$, non-zero $\langle P1 \rangle$, first order increase in density at T_{N_S-N} , multiple low-angle X-ray peaks) and as such are diagnostic for the splay nematic phase. Our work shows these behaviours to be intimately related to polar order in liquids.

We note that the timescale we use is almost an order of magnitude longer than that used in a previous study of RM734 (Ref. 18) by Clark et al., and performed on a simulation that is almost twice as large in terms of number of molecules.

Reviewer #3 (Remarks to the Author):

Accept, subject to corrections.

The discovery of the N_s phase by the authors is one of the most important made in recent few years in the subject of liquid crystals. Following the discovery of the Twist-Bend Nematic phase at the start of the decade, interest in alternative nematic phases has been intense; arguably the most important discovery that could be made in this field is that of a ferroelectric nematic, which is what is reported by the current authors work. The authors have made a number of prior publications on this topic, and so it is important to validate whether or not this paper is of sufficient new importance in its own right, to make it suitable for Nature Communications.

The current paper seeks to do a very detailed study of two similar materials: the first found N_s material RM734, and an equivalent molecule with a terminal CN rather than terminal NO₂ (ie RM734CN). It compares the various relevant physical properties (the measurements for the RM734CN are all new) and importantly an in-depth comparison of the Molecular Dynamics for the two structures. I believe that this level of detail is exactly what is required and is a most important question. This makes the subject of the paper and the presented results certainly worthy of publication in Nature Communications.

I have some reservations on whether or not the text as it stands does sufficient justice to the results, and believe that the authors should consider the following general comments:

1: The behaviour of RM734CN is far more unusual than the authors give credit. That is, unlike a conventional nematic, it shows a strong collective dipole mode (though weaker than the Ns material it is still exceptionally strong compared to conventional nematics); it shows the same unusually low splay elasticity (although it does not show the critical decrease that occurs immediately above the Ns transition which it does not exhibit in the temperature range that is accessible through the hysteresis before freezing to the crystal form.

We fully agree with the referee. Comparison with RM734 might hinder the relevance of the findings for RM734-CN and our wording might have contributed in this respect. The presence of a collective mode and the low value of the splay elastic constant do are not behaviours found in usual nematic phases. The extraordinary character of these findings has been stressed in the main text. More detailed answer in this respect can be found below, following the detailed comments of the referee.

<<ORIGINAL TEXT>>

Results:

On the other hand, $\Delta\varepsilon_{||,1}$ and $\Delta\varepsilon_{||,2}$ in RM734-CN are comparable, indicating that collective reorientations, although present, are weak.

<<NEW TEXT>>

Results:

On the other hand, $\Delta\varepsilon_{||,1}$ and $\Delta\varepsilon_{||,2}$ in RM734-CN are comparable, indicating that collective reorientations, although present, are weaker. That being said, the appearance of a developing collective mode in the N phase and with $\Delta\varepsilon_{||,1}$ reaching values up to 60, is far from being usual.

Added to the Discussion at the end of first paragraph:

Finally, it should be stressed that comparison with RM734 might hinder the relevance of the findings for RM734-CN, however behaviour of RM734-CN is far from being that of usual N phases. Splay elastic constant for RM734-CN is particularly low when compared with usual nematic materials as 5CB. Moreover, the presence two low frequency relaxation modes, one arising from collective motions and whose strength reaches values up to 60 clearly differs from the distinctive dielectric spectra of traditional nematic phases formed by rod-like molecules. For the latter, low frequency spectra is governed by a single molecular relaxation mode associated to the reorientation of the molecular around their short axis.

2: Given the importance of the results, I believe that the paper would be greatly improved by including data for RM551 (a second Ns material included right at the end of the paper) and a compound that is still further removed from the polar behaviour than RM734CN (I imagine RM734CN without the Methoxy side group should be sufficiently non-wedge shaped). That is, there needs to be validation of the Nd behaviour throughout the paper, rather than just at the end. There also needs to be clarification that the unusual behaviour observed by the RM734CN material (even though it does not show an Ns phase across the accessible temperature range) is at the borderline to Ns formation. That is, to show that the highly polar nature of RM734CN does not occur when its wedge shape is reduced (through removal of the transverse methoxy, or much longer terminal alkoxy group but without the fluorination used in RM551, etc) rather than the changes associated with the terminal dipole alone.

This is a sound suggestion - we are in agreement with the referee that these inclusions have offered us an opportunity to strengthen the manuscript significantly beyond the original submission. We have performed extensive new MD simulations on three materials that satisfy these criteria (fluorination in RM554, terminal chain length in RM500, lack of transverse methoxy in RM63). Of these materials, only

RM554 exhibits the N_s phase, the others being only nematogenic; their synthesis and characterisation was reported by us (Mandle) previously. The results we obtain from simulations are in agreement with points in the manuscript already, namely that the occurrence of the N_s phase is driven by enhanced packing in the polar state (versus the apolar).

We have added the following text:

<< NEW TEXT >>

Fig. 8: Additional structural variants of *RM734* explored by MD.

Thusfar we have restricted our atomistic MD simulations to just *RM734* and *RM734CN*, concluding with the hypothesis that the onset of polar order and the splay nematic phase is driven by enhanced packing. We can further support this argument by simulating additional variants of *RM734*, shown in Figure 8, in polar and apolar nematic configurations and looking for characteristic increased packing. We chose two materials which do not exhibit the N_s phase: *RM63* lacks the lateral methoxy group, whereas *RM500* has a longer terminal alkyl chain. The synthesis and mesomorphic behaviour of both was reported previously⁹. We also studied an additional material, *RM554*, which exhibits the N_s phase, and shows the same distinct X-Ray scattering patterns as *RM734* (Supplementary Figure 18). *RM554*, possesses an additional fluoro substituent adjacent to the nitro group⁹.

	P_2	B	$P_1(n)$	P_1 (dipoles)	P (C/m^2)	ρ ($g\ cm^3$)
RM63 (polar)	0.76 ± 0.010	0.017 ± 0.009	0.85 ± 0.06	0.83 ± 0.04	0.051 ± 0.002	1.260 ± 0.003
RM63 (apolar)	0.62 ± 0.021	0.026 ± 0.014	0.092 ± 0.009	0.068 ± 0.006	0.005 ± 0.0004	1.259 ± 0.003
RM500 (polar)	0.67 ± 0.034	0.064 ± 0.020	0.86 ± 0.08	0.84 ± 0.03	0.049 ± 0.003	1.175 ± 0.003
RM500 (apolar)	0.64 ± 0.024	0.073 ± 0.021	0.045 ± 0.011	0.022 ± 0.009	0.002 ± 0.0004	1.179 ± 0.003
RM554 (polar)	0.66 ± 0.015	0.027 ± 0.014	0.88 ± 0.09	0.87 ± 0.07	0.069 ± 0.0011	1.324 ± 0.003
RM554 (apolar)	0.68 ± 0.009	0.024 ± 0.008	0.061 ± 0.021	0.031 ± 0.011	0.009 ± 0.0007	1.311 ± 0.003

Table 2: Calculated order parameters and density for the *RM734* variants. Second-rank orientational order parameter (P_2), biaxial order parameter (B), polar order parameter (P_1), order parameter of the polarization vector (P_1 (dipoles)), polarization vector (P) at 400K for polar or apolar molecular dynamics simulations of homologous of *RM734*; all values are an average over each time step in the production MD run (30 – 280 ns) as described in the text, with plus/minus values corresponding to one standard deviation from the mean.

All new simulations exhibit nematic order, as shown by the values of P_2 , and the biaxial order parameter is close to zero for all (Table 2). Whereas polar simulations give large values of P_1 , P_1 (dipoles), and polarization vector P , these take near-zero values in the apolar simulations. The calculated density presents an important differentiation between materials that exhibit the N_s phase (*RM554*) and those that do not (*RM63*, *RM500*). If we make the argument that the splay-nematic phase is observed when the locally polar nematic configuration has a higher density than the equivalent apolar configuration, we begin to understand the molecular origins of this mesophase. For *RM63* the calculated densities are effectively identical, and for *RM500* the polar nematic is slightly lower in density than the apolar configuration. These results show that a lateral group is essential to give the packing advantage enjoyed by the polar configuration, however the presence of long terminal chains nullifies this advantage. These confinements to molecular structure present some impediments to the future engineering of

materials for applications utilising the N_s phase, which will presumably operate at ambient temperatures, however this goes beyond the scope of this paper.

The calculated density of $RM554$ is significantly higher in the polar configuration than in the apolar nematic (~0.9%), which mirrors the behaviour seen for $RM734$. Notably, the increased dipole moment of the fluoro-substituted $RM554$, relative to the parent $RM734$, leads to a larger calculated polarisation vector (P) and, one expects, would lead to a larger measured spontaneous polarisation (P_s). This presents a route to future tuning of spontaneous polarisation of N_s materials, with obvious ramifications for future applications. Pair-correlation analysis of an MD simulation of $RM554$ in the polar nematic configuration reveals the same head-to-tail packing mode observed for $RM734$ (Supplementary Figure 18). Altogether, these observations highlight the possibility of using MD simulations as a predictive tool for molecular design; by comparing densities of polar and non-polar configurations, through analysis of pair-correlation functions, in the pursuit of materials showing the polar nematic phase in the desired temperature range. GPU accelerated MD packages, and widespread availability of hardware, enable simulations with >100 ns day⁻¹ of performance for simulations featuring several tens of thousands of atoms (or more) that are required for stable liquid crystalline order to emerge, raising the possibility of N_s materials design guided by simulation.

A logical question to ask is why the polar and ferroelectric splay nematic phase had not been observed previously given that it should be ubiquitous in rod-like molecules with large transverse electric polarity. This is more puzzling when we consider that such materials are widely employed in display technology, being studied *en masse* for over half a century. For display applications, materials must operate at ambient temperatures – this is typical achieved *via* mixture formulation of materials incorporating long (n-C₃H₇ to n-C₉H₁₉) terminal alkyl chains; such structural features are incompatible with the N_s phase, as it is currently understood. Furthermore, in the context of materials for nematic LCDs, nitro substituents are generally inferior to nitriles, which themselves have been supplanted by multiply fluorinated materials.²⁷ At first glance it may appear unusual that the N_s phase had not been encountered until recently, but there is in fact far less overlap between the types of molecule studied so widely for display technology and those that exhibit this new nematic variant than is first apparent.

<< END NEW TEXT ADDITION >>

3. It would be harsh to condemn the paper to a lesser journal if the authors do not already have the results that show more conventional behaviour from a larger molecular deviation than $RM734CN$ alone. The authors have correctly found a very close analogue in that material, and therefore how sensitive the formation of the phase is. The fact that the phase is also shown in $RM551$ (as are the differences recorded in table 1) is . However, if they cannot add such results from a molecular variant that is FURTHER from the formation of the N_s phase than $RM734CN$ they should at least include some comments on this in the discussion.

As noted above, we have explored additional materials by MD. The choice of these materials was made based on the fact that they have been characterised experimentally and are ‘one variation’ in structure away from $RM734$ or its progenitor $RM230$. Experimental findings of these newly studied variants were previously reported in Mandle, R. J., et al. *Physical Chemistry Chemical Physics* 19, 11429–11435 (2017) and Mandle, R. J., et al., *Chemistry - A European Journal* 23, 14554–14562 (2017).

4. Although it is not clear from the text, the results are largely from nematic and N_s phases supercooled below their melting temperatures. It is not clear from the description for the reader, whether or not the MD simulations suggest that $RM734CN$ would never form the N_s phase even if supercooled to a much lower temperature than is possible practically due to crystallisation, or whether it would eventually form the N_s phase at some much lower temperature. This latter possibility seems precluded by the fact that the the N_s phase is quenched by just a 10% mixture of the $RM734CN$, but it is not clear why this effect would be so strong from the MD.

We make the argument from MD that an increased density in the polar configuration (versus the apolar) is an indicator that a material will exhibit the N_s phase under laboratory conditions. In our variable temperature MD simulations we never observe this divergence for $RM734CN$, which suggests that this material will never form the N_s phase in its neat state, or at least not at these temperatures. Given that cooling below 250K, the lowest temperature we study in MD, will almost certainly lead to crystallisation or glassification of $RM734CN$ we do not expect this phase to be observed in this material.

5. The inclusion of $RM551$ was very powerful. It should be said at the outset that a model will be presented for the formation of the NS phase that is then tested successfully against a second material. (As I state above, I

would like to also see a second negative prediction too, for something less polar than the RM734CN, though that may not be possible).

I think in our original submission we perhaps missed the importance of this inclusion, and certainly, it is strengthened by the simulation and discussion of additional materials which were included at the suggestion of the referee. We hope that the inclusion of two more 'negatives', in RM500 and RM63, convince the referee of our argument; these are detailed elsewhere in this response.

We have also added a statement in the introduction that a model is presented and then tested against further materials (new text in **bold**):

"The differences between both materials are then analysed in-depth *via* molecular dynamics simulations; **enabling us to develop a predictive model which is then successfully tested against three further variants of RM734, one of which also exhibits the N_s phase.**"

Below are comments written during the first reading of the text that may be apposite. Some of these may already have been covered above.

The introduction states that it is important to understand the molecular design rules, and the paper concentrates on swapping the terminal nitro for a terminal cyano moiety. However, the materials both use methoxy terminated mesogens, which inherently will lead to higher crystallisation temperatures. At the end of the paper, the authors include a longer terminal alkyl group from their prior art, and found the methoxy also essential for the formation of the N_s phase. However, some comment on this would be helpful for the reader in the introductory remarks.

I suppose it is fair to say that understanding these molecular design rules is an ongoing problem, and the subject of this paper. I think it is also important to say we are not 'holding anything back'; the contained results are very much the state of the art on our behalf, and given the probable applications that N_s materials will find, we are eager to learn much more. We have added the following text:

So far, it has been shown that for RM734-like materials, a short terminal chain (OMe, OEt) coupled with a lateral group (OMe, OEt, or OⁿPr) are prerequisite for the formation of the N_s phase, as is a terminal nitro group. The reasons for the dependency of the ferroelectric N_s phase upon these structural features are not clear; understanding them is clearly a significant barrier to developing improved materials, which may eventually be deployed into applications.

Figure 2. The caption should make clear that RM734CN was used for a). It would also be useful to include similar results to these for RM734. These are found by going to the SI of reference [3]. Repeating in the present SI would be beneficial.

Caption for Figure 2 was clarified and the label for the material was added also in the graph. We agree with the referee that inclusion of data and fit examples for RM734 can be beneficial for the readability and clarity of the presented results, and thus have been added to the supporting information.

Supplementary Figure 5. Examples at different temperatures of the measured dielectric spectra and their fits for RM734. (left) Frequency dependence of the real (full circles) and imaginary (open circles) dielectric permittivity. Solid lines result from fitting to Equation 1 in the manuscript and the corresponding deconvolution into the elementary processes. Dashed lines correspond to the current conductivity term. (right) The derivative of the real part of the permittivity $d(\epsilon')/d\log(f)$ at the corresponding temperatures allows for better visualization of the relaxation modes and benevolence of the fits. Modes $m_{||,1}$ and $m_{||,2}$ overlap in frequency at high temperatures and is not possible to resolve them above 160 °C. For those high temperatures only one mode was considered in the fitting, where the amplitude of the lower frequency mode $m_{||,1}$ makes it prevail. Source of data and fits: Sebastian et al., PRL, 124 (3), 037801 (2020).

The results shown here for RM734CN are ambiguous. This is because the crystallisation point for this material is quoted as 173°C with no mention of hysteresis on heating and cooling.

It is true that the original text was not clear and we thank the referee for the help on clarifying this point. We had amended the text to this effect in the Materials section in Methods and complemented it with Supplementary Figure 1 and Supplementary Figure 2 showing polarizing optical microscopy images. For detailed description of the added text, please refer to response below (to the comment “The minimum temperature for....”)

The text describing the diagram suggests that the 1-mode relaxation is Arrhenius across the nematic phase, but no results are shown on the figure for this: only for the 2-mode. There are results for all 3 relaxations 1,2,3 from 120°C to 172°C. Are these from a supercooled nematic phase, or measured in the crystal phase? Alternatively, the statement in the text that $m||,1$ relaxation frequency of RM734CN is Arrhenius in its nematic phase might better apply to $m||,2$; this is the only relaxation shown in the range 172°C - 200°C that corresponds to the nematic temperature range. I presume that this lack of clarity could be corrected by a simple statement earlier that the samples supercooled in the nematic phase to below 120°C.

As just stated above, on cooling, N phase is supercooled down to 115 °C, temperature at which sample started crystallizing. At low temperatures, below 172 °C, two modes are clearly resolved (namely $m||,1$ and $m||,2$) in the spectra. However, approaching that temperature the deconvolution of the data into two different modes is impossible (or highly ambiguous) because of the large overlap. In this case, to avoid making assumptions and show only what the data safely allows to conclude, we only used one (broader) mode for the fit in that high temperature region (red circles in Figure 2 between 172 °C and 200 °C). The temperature at which the fit strategy changes from one to two relaxation modes was taken by the temperature at which the derivative of the real part (Fig.SI.4) unambiguously reveals two modes. The same was done for RM734 as it can be seen in the ‘splitting’ at around 160 °C.

We imply that both modes are present up to the isotropic phase (both with Arrhenius behaviour), but we are just not able to separately deconvolute them unambiguously.

The manuscript text has been updated to clarify this point and due to the word limits of the manuscript, a more detailed description of the fit strategy was also added in the Supplementary Note 2.

The minimum temperature for obtaining the Ns phase in RM734 would also be instructive, though the results shown in Fig 2 seem to only go to 130°C, which is only 2-3°C into the Ns phase.

This confusion evidences that our previous draft was not clear enough with respect to the phase behaviour of both materials. Consequently, the Materials subsection in Methods has been greatly extended to include a thorough description of the phase behaviour of both materials and new supplementary figures have been added, showing phase behaviour as observed by polarizing optical microscopy at different conditions.

<<NEW TEXT>>

Both RM734 and RM734-CN were synthesised via literature methods⁹; their chemical structures and transition temperatures (°C) are given in Figure 1, with monotropic phase transitions as reported by DSC measurements are presented in parenthesis (⁹). All the experiments presented here have been performed on cooling. On cooling, RM734-CN shows an isotropic-nematic transition around 200 °C. However, the nematic phase can then be supercooled to temperatures around 120 °C even at cooling rates as slow as 0.25 °C/min.

On the other hand, RM734 shows a very rich phase behaviour. On heating from room temperature, a Cr-Cr (Cr I to Cr II) transition is observed between 80-90 °C depending on the heating rate (Supplementary Figure 1). On further heating, the Cr II directly melts into the *N* phase around 140 °C. Now, without heating further into the isotropic phase, cooling the sample reveals the following phase transition: *N* – 132.7 °C – *N_s* – 83 °C – Cr I (Supplementary Figure 1b). The exact temperature at which the sample crystallizes depends on the cooling rate. Interestingly, depending on the history of the sample, the cooling rate and the cell surfaces, *N_s* phase can be maintain down to room temperature (Supplementary Figure 2) or a third Crystal phase can be obtained (Cr III) (Supplementary Figure 1.c). Subsequent heating from Cr III reveals that the *N_s* phase can be obtained also on heating, which then transitions into the *N* phase around 130 °C. During this transition some aligning features of the *N_s* phase are retained which give rise to a further anchoring relaxation in the *N* phase. Details on these structural features are given elsewhere¹⁷. (<https://arxiv.org/abs/2103.10215>)

<<END OF NEW TEXT>>

<<ADDITION TO SUPPLEMENTARY INFORMATION>>

Polarizing optical microscopy experiments were performed in an Optiphot-2 POL Nikon microscope. Images were recorded with a Canon Eos 100D camera. The sample was held in a heating stage (Instec HCS412W) together with a temperature controller (mK2000, Instec).

Supplementary Figure 1. Phase behaviour of RM734 as investigated by polarizing optical microscopy. (a) Textures observed in an EHC 8 μm thick planar cell when heating at 5 °C/min from room temperature. (b) Phase sequence observed for the same cell when cooling from the *N* phase down to room temperature at 2 °C/min. (c) Under certain conditions (thermal history and cell surfaces) a third crystalline phase (Cr-III) can be obtained when cooling from the *N_s* phase. Here we show for example, textures observed for an Instec IPS cell with antiparallel rubbing directed parallel to the electrodes and a thickness of 9 μm . When heating, Cr-III melts directly into the *N_s* phase and *N_s*-*N* transition is observed around 130 °C. Domain structures shown in (c) are discussed elsewhere (<https://arxiv.org/abs/2103.10215>)

Supplementary Figure 2. Polarizing optical microscopy image of RM734 at room temperature after fast cooling from the N phase, showing that under certain conditions Ns phase can be supercooled down to room temperature.

<<END ADDITION TO THE SUPPORTING INFORMATION>>

The dielectric results of RM734CN in Fig 2 show a polar, collective relaxation $m_{||,1}$ in the crystal (or supercooled N) phase. It is stated that this is weak, but it is surprisingly strong - with a dielectric strength of 10. This is very large indeed if compared to say similar nematic phases (eg 5CB) that shows no collective motion. It is about half the magnitude of the higher frequency dielectric relaxation of the longitudinal dipole moment $m_{||,2}$. Given this, it would be apposite to make some comment in this section that it is still high even though no Ns phase was found for this material, rather than just stating that it is much weaker than for the compound RM734 that does show the Ns phase. That is, arguably, it is the similarity between the materials (low K11, pre-transitional collective dielectric mode) that unusual given that the material does not form an Ns phase.

As stated above, $m_{||,1}$ and $m_{||,2}$ are too close in frequency at high temperatures and are fitted as a single mode with amplitude increasing after the I-N transition up to strength values around 40. When the fit criteria is changed to two relaxation processes, such value is then divided between the strengths of both modes $m_{||,1}$ and $m_{||,2}$, where initially $m_{||,2}$ prevails. On further cooling, strength of the collective $m_{||,1}$ slowly increases and surpasses that of $m_{||,2}$. Values vary then between 21 and 42 for $m_{||,2}$ and between 13 and 52 for $m_{||,1}$.

We agree with the referee that this behaviour is far from being similar to that of a classical N liquid crystal like 5CB. 5CB has a dipole moment around 4.5-5D and dielectric relaxation associated to the rotation around the short molecular axis has an amplitude around 10. Considering that the dipole moment for RM734 and RM734-CN is approximately 2 times that of 5CB it is not surprising that the strength of $m_{||,2}$ lies in the range described before, moreover taking into account that the overlap between both modes makes it difficult to resolve them precisely. However, what is special is the appearance of the lower frequency collective mode, which is not expected for a classical N phase. The onset of the collective behaviour is there for RM734-CN. Comparison with RM734 might make it look negligible, but as the referee states, it is surprisingly strong when compared with classical N phases.

Text in the main manuscript has been amended to include these suggestions. See response to point 1 of Referee 3.

For RM734, only the pretransitional behaviour for the Ns phase is shown, and there are no results presented for below the N-Ns transition. Some comment on why this should be included here (presumably it appeared in ref [3]). Presumably (given fig 3) the relaxation becomes very low frequency as the viscosity diverges.

The referee is right, when approaching the N-Ns transition relaxation frequency of the collective mode becomes very low and the mode is masked by conductivity effects. We have measured dielectric spectra

in the Ns phase. The analysis and interpretation of such data is not straightforward at all and, indeed, it is still ongoing work which is out of the scope of this paper. RM734 shows an extraordinary large value of spontaneous polarization ($6 \mu\text{C}/\text{cm}^2$) but it also has large conductivity contribution. Both effects together can lead to multiple effects that must be carefully considered before reaching any conclusions.

Again on the subject of how SIMILAR RM734 and RM734CN are, both have unusually low K11. The difference is that there is a critical pretransitional effect in RM734 above Ns-N but this occurs over a narrow temperature range. One can envisage that RM734CN might super cool to a sufficiently low temperature and then form the Ns phase and that critical behaviour then be observed. That is, it is not clear from the paper whether or not RM734CN could exhibit the Ns phase at some temperature, or whether its configuration prevents this from ever happening. The discussion begins with discussion that seem to preclude this possibility, but it is not clear how that conclusion could be made.

As stated above we make the argument from MD that an increased density in the polar configuration (versus the apolar) is an indicator that a material will exhibit the Ns phase under laboratory conditions. The new, clearer, representation of simulation densities introduced in the new Fig. 5 allows us to substantiate on this point. As seen in Fig. 5d (shown below for readability) we simulated density down to 250K, which in the case of RM734-CN corresponds to 140 K lower temperature than the minimum N temperature accessible by experiments before crystallization. What new Fig 5.d shows is that there is no difference in the simulated density for the polar and apolar configurations in RM734-CN, even at lower temperatures that those studied experimentally. Thus, we expect that the RM734-CN will remain, even at sufficiently low temperatures, in the N phase as entropically it is more favourable.

Page 7 - dipole vectors need units.

Units were added in the main text.

SI

Spelling of "consequently".

Fig S1-2 points are mislabelled as squares and circles rather than open and closed points.

We thank the referee for spotting the typo and the mislabelling. We corrected the typo and the caption. "Frequency dependence of the real (full circles) and imaginary (open circles) dielectric permittivity."

REVIEWERS' COMMENTS

Reviewer #1 (Remarks to the Author):

The questions raised by the reviewers have been punctually addressed. The paper has been thoroughly revised and the new version is greatly improved.

I am not fully convinced of the explanation that is proposed for the origin of the polar nematic phase, but this may be a personal opinion.

Considering that this paper presents novel results on a problem of current interest, I think that it could be published in Nature Communications with some minor modifications, as suggested below.

- Positional and orientational correlations

Sentences like the following:

"The implication being that the presence of multiple low angle scattering peaks is a consequence of polar nematic order rather than differences in molecular structure"

"The additional small angle X-ray scattering peaks observed for splay-nematic materials such as RM734 appear to be a consequence of the unique dipole ordering"

remain rather generic.

I wonder if, based on the MD simulations, an effort can be made to substantiate them (e.g. is it possible to more precisely relate the small angle peaks observed in the presence of polar order to structural features of the polar organization?)

Figures 5e-h are very small and it is hard to distinguish their content and to follow the comments in the text; larger images would be useful. Moreover, Figures 5e and 5f seem to be different from the corresponding figures in Ref. 18 (although it is difficult to say, given the size of the images); I wonder if a comment on this could be added.

Finally I suggest to introduce a comment on the RDF in Supplementary Figure 14 and Supplementary Figure 15 (e.g. what are the dashed lines in the plots?) Also a couple of words to stress the difference between the RDF and the pair correlation function could be useful to readers.

- Diffusion coefficient

I haven't found an explanation of how diffusion coefficients are calculated.

Moreover, it might be worth distinguishing longitudinal and transversal components of the diffusion tensor, as customary for nematics.

- Typos and other minor points

line 416 "can be maintain down to"

line 465 "Compressabilities"

Caption to Supplementary Figure 8: "Direction of the inertia tensor"

The concept of direction of a tensor is not clear to me; I suppose this is the principal axis of the inertia

tensor, i.e. the 'mass inertia axis' defined in the main text (line 160).

Throughout the paper there are some obvious comments, that in my opinion could be removed without any detriment, e.g.

Lines 300-304 "As we have shown, taken on their own, electronic structure calculations of isolated molecules and molecular conformational potential energy surfaces, cannot offer a plausible explanation for the formation of the splay nematic phase and its polar nematic order. Contrarily, the results of molecular dynamics simulations offer several key observations."

Reviewer #2 (Remarks to the Author):

The revised manuscript has been properly strengthened by the authors and has become even more convincing. The molecular origin proposed in this paper has been studied only in a limited number of materials, so it is not a highly general understanding, but it could be a great step toward a universal understanding of the ferroelectric nematic phase. I believe that this work will be of interest to a wider audience across disciplines.

Reviewer #3 (Remarks to the Author):

Recommendation: Publish with minor corrections.

The authors have undergone extensive modifications to the paper and its Supplementary Material. I believe that they have either made the requested changes asked by all three referees, or have justified satisfactorily the reasons for not making changes (in most cases). In particular the paper now has a far more detailed analysis following the approach of ref 18, and has included modelling not only for a second compound that has the Ns phase but also other negative examples. In this it is much more complete and suitable for publication in Nat. Comm.

Minor points that can be addressed at the editing stage:

Values of h should include their units (Angstrom) where omitted in the text. I may be technically okay to state $h = 0$ without units, but certainly not for $h = 20$. Similarly, the new text also mentions $P1$ and $P2$ which are values of the Legendre polynomials, rather than their mean values over the chosen ensemble. Surely, $\langle P1 \rangle$ and $\langle P2 \rangle$ should be used instead.

The comparison of order parameters with ref [18] in the new text on page 11 needs the uncertainties being quoted too (ie $\langle P2 \rangle = 0.787 \pm 0.009$ and $\langle P1 \rangle = 0.924 \pm 0.003$). Thus, strictly speaking both $\langle P2 \rangle$ and $\langle P1 \rangle$ are both lower in this new work, although the reasons given should still apply.

New text on page 4. With the removal of the Havriliak-Negami equation, the rather colloquial use of the "Debye" to mean a "Debye-type relaxation" is now a little too confusing. It is better to use the phrase "Debye-type relaxation" instead.

We thank the Editor and reviewers for their prompt consideration of this manuscript. Thank you for your thorough review, comments and suggestions. We have sequentially addressed the comments point by point below and we offer this response.

REVIEWERS' COMMENTS

Reviewer #1 (Remarks to the Author):

The questions raised by the reviewers have been punctually addressed. The paper has been thoroughly revised and the new version is greatly improved.

I am not fully convinced of the explanation that is proposed for the origin of the polar nematic phase, but this may be a personal opinion.

Considering that this paper presents novel results on a problem of current interest, I think that it could be published in Nature Communications with some minor modifications, as suggested below.

- Positional and orientational correlations.

Sentences like the following:

“The implication being that the presence of multiple low angle scattering peaks is a consequence of polar nematic order rather than differences in molecular structure”

“The additional small angle X-ray scattering peaks observed for splay-nematic materials such as RM734 appear to be a consequence of the unique dipole ordering”

remain rather generic.

I wonder if, based on the MD simulations, an effort can be made to substantiate them (e.g. is it possible to more precisely relate the small angle peaks observed in the presence of polar order to structural features of the polar organization?)

The low angle peaks are rather diffuse and broad, which is probably owing to their resulting from many different species of interacting molecules (i.e. n-mers). Unfortunately, it is not clear to us that we could narrow this information down into a dataset which is communicable. We would be limited to selecting a few examples, as we did for the RDF analysis in the SI.

Figures 5e-h are very small and it is hard to distinguish their content and to follow the comments in the text; larger images would be useful. Moreover, Figures 5e and 5f seem to be different from the corresponding figures in Ref. 18 (although it is difficult to say, given the size of the images); I wonder if a comment on this could be added.

We agree that Figure 5e-h was too small to be of use and so we have included a revised version of this figure, in which these elements are larger.

Superficially, there are small visual differences between our work and ref 18 – they may be a consequence of us running simulations that are approx. 12x longer, or they may be due to the different plotting used, or the use of different bin widths when computing the pair correlation function.

Finally I suggest to introduce a comment on the RDF in Supplementary Figure 14 and Supplementary Figure

15 (e.g. what are the dashed lines in the plots?) Also a couple of words to stress the difference between the RDF and the pair correlation function could be useful to readers.

We agree some comment on the difference between RDF and pair correlation functions could be added, we have added the following text in Supplementary Note 8:

“The RDF presented here are isotropic, being calculated for spherical shells, and are calculated between specific sets of atoms corresponding to functional groups within a given molecule. On the other hand, the pair correlation functions (PCF) presented within the manuscript are anisotropic, being calculated for cylindrical shells oriented with their length along the nematic director, and are computed between the centres-of-mass of all molecules within the simulation. “

Regarding the dashed lines, this is detailed in the figure captions in the SI already – as these RDFs are between atoms (rather than centre-of-mass) we find very strong intramolecular peaks. These intramolecular peaks are of little relevance to our analysis, so we use a dashed plotting style to keep the readers focus on the pertinent data. As this part of the text is in the SI and not the manuscript body, we can afford some extra words to elaborate on this:

“The intermolecular RDF between given sets of atoms is presented as a solid line, whereas the dashed line corresponds to the intramolecular RDF for the same set of atoms.”

- *Diffusion coefficient*

I haven't found an explanation of how diffusion coefficients are calculated.

Moreover, it might be worth distinguishing longitudinal and transversal components of the diffusion tensor, as customary for nematics.

This is a sound suggestion. Description of the methodology was included in the Supplementary Note 8 alongside the obtained values:

“Molecular diffusion coefficients were calculated from the MD simulation trajectories for RM734 and RM734-CN in both polar and apolar states using the Gromacs tool gmx msd as supplied with Gromacs/2019.2. “

Calculation of longitudinal and transverse components of the diffusion tensor is a good suggestion, but it is only possible to obtain scalar values from our version of Gromacs. Although this requires us to develop new code/software tools, we will look to revisit this in future, and we thank the referee for a very interesting suggestion.

- *Typos and other minor points*

line 416 "can be maintain down to"

line 465 "Compressabilities"

We thank the referee for spotting the typos, which we now corrected.

Caption to Supplementary Figure 8: "Direction of the inertia tensor"

The concept of direction of a tensor is not clear to me; I suppose this is the principal axis of the inertia tensor, i.e. the 'mass inertia axis' defined in the main text (line 160).

We thank the referee for spotting this point and helping us. Caption has been corrected.

Original text:

Directions of *the inertia tensor* and dipole moment vector for RM734 and RM734-CN as calculated using M06HF-D3/aug-cc-pVTZ level of DFT.

New text:

Directions of the *mass inertia axis* and dipole moment vector for RM734 and RM734-CN as calculated using M06HF-D3/aug-cc-pVTZ level of DFT.

Throughout the paper there are some obvious comments, that in my opinion could be removed without any detriment, e.g.

Lines 300-304 "As we have shown, taken on their own, electronic structure calculations of isolated molecules and molecular conformational potential energy surfaces, cannot offer a plausible explanation for the formation of the splay nematic phase and its polar nematic order. Contrarily, the results of molecular dynamics simulations offer several key observations."

We agree with the referee and the comment has been deleted

Reviewer #2 (Remarks to the Author):

The revised manuscript has been properly strengthened by the authors and has become even more convincing. The molecular origin proposed in this paper has been studied only in a limited number of materials, so it is not a highly general understanding, but it could be a great step toward a universal understanding of the ferroelectric nematic phase. I believe that this work will be of interest to a wider audience across disciplines.

We thank the referee for the supportive comments.

Reviewer #3 (Remarks to the Author):

Recommendation: Publish with minor corrections.

The authors have undergone extensive modifications to the paper and its Supplementary Material. I believe that they have either made the requested changes asked by all three referees, or have justified satisfactorily the reasons for not making changes (in most cases). In particular the paper now has a far more detailed analysis following the approach of ref 18, and has included modelling not only for a second compound that has the Ns phase but also other negative examples. In this it is much more complete and suitable for publication in Nat. Comm.

Minor points that can be addressed at the editing stage:

Values of h should include their units (Angstrom) where omitted in the text. I may be technically okay to state $h = 0$ without units, but certainly not for $h = 20$.

We thank the referee for spotting this mistake, which we now corrected.

Similarly, the new text also mentions $P1$ and $P2$ which are values of the Legendre polynomials, rather than their mean values over the chosen ensemble. Surely, and should be used instead.

We have defined both $P1$ and $P2$ in equations (1) and (2) in Methods respectively; according to our definitions they are mean values over the ensemble.

The comparison of order parameters with ref [18] in the new text on page 11 needs the uncertainties being quoted too (ie $= 0.787 \pm 0.009$ and $= 0.924 \pm 0.003$). Thus, strictly speaking both are both lower in this new work, although the reasons given should still apply.

We added the corresponding uncertainties from reference 18.

New text on page 4. With the removal of the Havriliak-Negami equation, the rather colloquial use of the "Debye" to mean a "Debye-type relaxation" is now a little too confusing. It is better to use the phrase "Debye-type relaxation" instead.

We fully agree with the referee, Debye-type relaxation has been now used instead of Debye.